Manuscript prepared for Clim. Past
with version 2014/09/16 7.15 Copernicus papers of the LaTeX class copernicus.cls.
Date: 28 September 2020

# The middle-to-late Eocene greenhouse climate, modelled using the CESM 1.0.5

Michiel Baatsen[1], Anna S. von der Heydt[1], Matthew Huber[2], Michael
A. Kliphuis[1], Peter K. Bijl[3], Appy Sluijs[3], and Henk A. Dijkstra[1]

[1]Institute for Marine and Atmospheric Research, Department of Physics, Utrecht University,
Princetonplein 5, 3584CC Utrecht, Netherlands
[2]Purdue University, 610 Purdue Mall, West Lafayette, IN, 47906 USA
[3]Department of Earth Sciences, Marine Palynology and Paleoceanography, Laboratory of
Palaeobotany and Palynology, Utrecht University, Princetonlaan 8a, 3584 CB Utrecht, the
Netherlands

*Correspondence to:* Michiel Baatsen (m.l.j.baatsen@uu.nl)

**Abstract.** The early and late Eocene have both been the subject of many modelling studies, but only
few have focused on the middle Eocene. The latter still holds many challenges for climate model-
lers, but is also key to understand the events leading towards the conditions needed for Antarctic
Glaciation at the Eocene-Oligocene transition. Here, we present the results of CMIP5-like coupled

climate simulations using the Community Earth System Model (CESM) version 1. Using a new
detailed 38Ma geography reconstruction and higher model resolution compared to most previous
modelling studies and sufficiently long equilibration times, these simulations will help to further
understand the middle-to-late Eocene climate. At realistic levels of atmospheric greenhouse gases,
the model is able to show an overall good agreement with proxy records and capture the important

aspects of a warm greenhouse climate during the Eocene.
With a quadrupling of pre-industrial concentrations (PIC) of both $CO_2$ and $CH_4$ (i.e. 1120ppm and
∼2700ppb), sea surface temperatures correspond well to the available late middle Eocene (42–
38 Ma; ∼Bartonian) proxies. Being generally cooler, the simulated climate under 2×PIC forcing
is a good analogue for that of the late Eocene (38–34 Ma; ∼Priabonian). Terrestrial temperature

proxies, although their geographical coverage is sparse, also indicate that the results presented here
are in agreement with the available information.
Our simulated middle-to-late Eocene climate has a reduced equator-to-pole temperature gradient and
a more symmetric meridional heat distribution compared to the pre-industrial reference. The collect-
ive effects of geography, vegetation and ice account for a global average 5–7 °C difference between

pre-industrial and 38Ma Eocene boundary conditions, with important contributions from cloud and
water vapour feedbacks. This helps to explain Eocene warmth in general, without the need for green-

house gas levels much higher than indicated by proxy estimates (i.e. $\sim$500–1200 ppm $CO_2$) or low latitude regions becoming unreasonably warm. High latitude warmth supports the idea of mostly ice-free polar regions, even at 2$\times$PIC, with Antarctica experiencing particularly warm summers. An overall wet climate is seen in the simulated Eocene climate, which a strongly monsoonal character. Equilibrium climate sensitivity is reduced (0.62 $^{\circ}$C/Wm$^{-2}$; 3.21$^{\circ}$C warming between 38Ma 2$\times$PIC and 4$\times$PIC) compared to that of the present-day climate (0.80 $^{\circ}$C/Wm$^{-2}$; 3.17$^{\circ}$C per $CO_2$ doubling). While the actual warming is similar, we see mainly a higher radiative forcing from the second PIC doubling. A more detailed analysis of energy fluxes shows that the regional radiative balance is mainly responsible to sustain a low meridional temperature gradient in the Eocene climate, as well as the polar amplification seen towards even warmer conditions. These model results may be useful to reconsider the drivers of Eocene warmth as well as the EOT, but can also be a base for more detailed comparisons to future proxy estimates.

## 1 Introduction

The Eocene-Oligocene transition (EOT) is one of the most dramatic climate transitions of the Cenozoic, thought to be associated with the formation of a continental-scale ice sheet on Antarctica (Zachos et al., 1994; Coxall et al., 2005; Lear et al., 2008). A possible cause for the inception of ice is a long-term decline of greenhouse gas concentrations through the middle Eocene, eventually crossing a threshold for glaciation (DeConto and Pollard, 2003; DeConto et al., 2008; Gasson et al., 2014). Following the early Eocene (~50Ma), a gradual cooling levelled off during the middle Eocene (43–42 Ma) and eventually reversed into a warming (Zachos et al., 2001, 2008; Bijl et al., 2009; Cramwinckel et al., 2018) towards the Middle Eocene Climatic Optimum at ~40Ma (MECO; Bohaty and Zachos 2003; Bijl et al. 2010; Sluijs et al. 2013). The cooling trend continued during the late Eocene (~38–34 Ma), with a cold interval at ~37.3Ma characterised by the Priabonian Oxygen isotope Maximum (PrOM, Scher et al. 2014). While these temperature changes may have caused some ice growth as early as the middle Eocene, they did not allow the formation of a continental-scale Antarctic ice sheet to occur until after 34Ma (Scher et al., 2014; Passchier et al., 2017; Carter et al., 2017). It remains a question to what extent continental geometry (e.g. opening of Southern Ocean Gateways) next to gradual shifts in both the atmospheric and oceanic circulation, was a driver to both regional and global climate change during the Eocene (Bijl et al., 2013; Bosboom et al., 2014; Goldner et al., 2014; Sijp et al., 2014, 2016; Toumoulin et al., 2020).

The climate throughout most of the Eocene was characterised by a reduced equator-to-pole temperature gradient compared to present day (Bijl et al., 2009; Hollis et al., 2012; Douglas et al., 2014; Evans et al., 2018). This aspect of the Eocene greenhouse climate has proven challenging to simulate adequately with global climate models (Huber and Sloan, 2001; Huber and Caballero, 2011; Cramwinckel et al., 2018). Very high greenhouse gas concentrations were often needed to reproduce high-latitude warmth at the expense of equatorial temperatures being significantly higher than indicated by proxy data (Huber and Caballero, 2011; Lunt et al., 2012). Estimates of the meridional temperature gradient during the Eocene have come up with respect to some earlier studies, mainly due to warmer equatorial temperatures (Pearson et al., 2007; Schouten et al., 2013; Inglis et al., 2015; Evans et al., 2018; Cramwinckel et al., 2018). Meanwhile, global climate models have been under continuous development by including more processes (especially cloud properties, e.g. Abbot et al. 2009; Kiehl and Shields 2013) and using a higher spatial resolution with better resolved palaeogeographies (Baatsen et al., 2016; Lunt et al., 2016; Hutchinson et al., 2018) to improve the overall model-proxy comparison.

Many modelling studies have focussed on the early Eocene (Huber and Caballero, 2011; Lunt et al., 2012; Herold et al., 2014; Zhu et al., 2019) or looked at the latest Eocene - early Oligocene (Hill et al. 2013; Ladant et al. 2014; Kennedy et al. 2015; Elsworth et al. 2017; Hutchinson et al. 2018; Kennedy-Asser et al. 2019, 2020; Tardif et al. 2020; see also the overview of Gasson et al. 2014).

However, there has been limited attention to the middle Eocene. Licht et al. (2014) performed simulations with both a 40Ma and 34Ma geography reconstruction, using a reduced complexity global climate model (FOAM). A set of comprehensive model studies with several time slices covering the Eocene (Inglis et al., 2015; Lunt et al., 2016; Farnsworth et al., 2019) use a lower resolution version of the HadCM3 model. Cramwinckel et al. (2018) provide an overview of different climatic states throughout the middle Eocene cooling using the CCSM3 simulations of Goldner et al. (2014).

Here, we present the results of a set of coupled atmosphere-ocean simulations with the Community Earth System Model (CESM) version 1.0.5 using 38Ma boundary conditions from Baatsen et al. (2016). The 38Ma boundary conditions differ from those used to study the EOT, primarily in the treatment of Southern Ocean gateways (i.e. narrow, shallow passages) and an open Turgai Strait. With time-specific geographic boundary conditions and adequate equilibration, the aim is to show a more detailed and representative model-based overview of the middle-to-late Eocene (i.e. Bartionian–Priabonian; ~42–34 Ma) climate. The considered period is suitable to investigate both the warm greenhouse climate of the middle Eocene and the conditions leading up to the EOT. The focus will therefore be on the general features of the modelled climate, a comparison to proxies as well as other model results, and the similarities/differences with the present-day climate (specifically regarding climate sensitivity).

The model set-up and experimental design of the CESM simulations are first explained in section 2. The main results are then presented in section 3 including the equilibrium climate of each simulation (3.1), a model-proxy comparison (3.2) and a model-model comparison (3.3). This is followed by an analysis of equilibrium climate sensitivity (3.4), with a focus on the main changes involved in the radiative balance. Finally, the main findings are summarised and discussed in section 4.

## 2 Methods

### 2.1 The CESM 1.0.5

The Community Earth System Model (CESM; Hurrell et al. 2013) version 1 is a fully coupled atmosphere-land-ice-ocean general circulation model (GCM) that was developed at the National Center for Atmospheric Research (NCAR) in Boulder, Colorado. For use in palaeoclimate modelling, version 1.0.5 of the CESM is a suitable choice motivated by a trade-off between increasing model complexity and computational cost. This version of the CESM is equivalent to the latest version of the Community Climate System Model (CCSM4; Blackmon et al. 2001; Gent et al. 2011). In the configuration used here, the horizontal resolution of both the atmosphere and ocean is doubled compared to that of Herold et al. (2014) while that of the atmosphere is about 50% higher than in Hutchinson et al. (2018), with comparable ocean grids.

The atmospheric component of the CESM is the Community Atmosphere Model (CAM4; Neale et al. 2013) which uses a finite volume grid at a nominal resolution of $2°$ ($2.5° \times 1.9°$) and 26 vertical levels with a hybrid sigma vertical coordinate extending upward to 2hPa. In this configuration, the model has a reported warming response of $3.13°C$ to a doubling of $CO_2$ starting from pre-industrial conditions (Bitz et al. 2012, compared to $\sim 2.5°C$ in CCSM3; Kiehl et al. 2006).

The physical, chemical and biological processes taking place on land are represented in the Community Land Model (CLM4; Oleson et al. 2010; Lawrence et al. 2011). A static rather than dynamic vegetation model is used here to avoid runaway feedback effects, which can become an issue especially in warm greenhouse climates (e.g. dieback of vegetation at low latitudes; Loptson et al. 2014; Herold et al. 2014). The considered biomes are translated into fractions of the corresponding CLM4 plant functional types (PFTs), from which a set of monthly forcing files is finally used in the model. Fresh water runoff is treated by a simple river routing scheme, in which all runoff is transported to one of the surrounding 8 model grid cells until the ocean is reached. The direction is determined by the local topography gradient and manually adjusted where the runoff scheme would otherwise form closed loops.

The sea ice component consists of the Los Alamos National Laboratory (LANL) Community Ice Code version 4 (CICE4; Hunke and Lipscomb 2008). For simplicity, sea ice only forms when the sea surface cools down to $-1.8°C$, after which its dynamical behaviour (e.g. melt and advection) is treated by the model specifically.

The CESM1 uses the LANL Parallel Ocean Program version 2 (POP2; Smith et al. 2010) for the ocean model component. The standard configuration is applied here, with a nominal $1°$ ($1.25° \times 0.9°$) horizontal resolution on a curvilinear grid placing the northern pole over Greenland. The POP2 model is set up with 60 layers of varying thickness between 10m near the surface and 250m at greater depth. Horizontal viscosity is considered anisotropic (Smith and McWilliams, 2003) and horizontal tracer diffusion follows the parameterisation of Gent and Mcwilliams (1990). The model further

uses the KPP-scheme to determine vertical mixing coefficients (Large et al., 1994). More information and discussion on the ocean model physics and parameterisations can be found in Danabasoglu et al. (2008, 2012).

## 2.2 Model Experiments

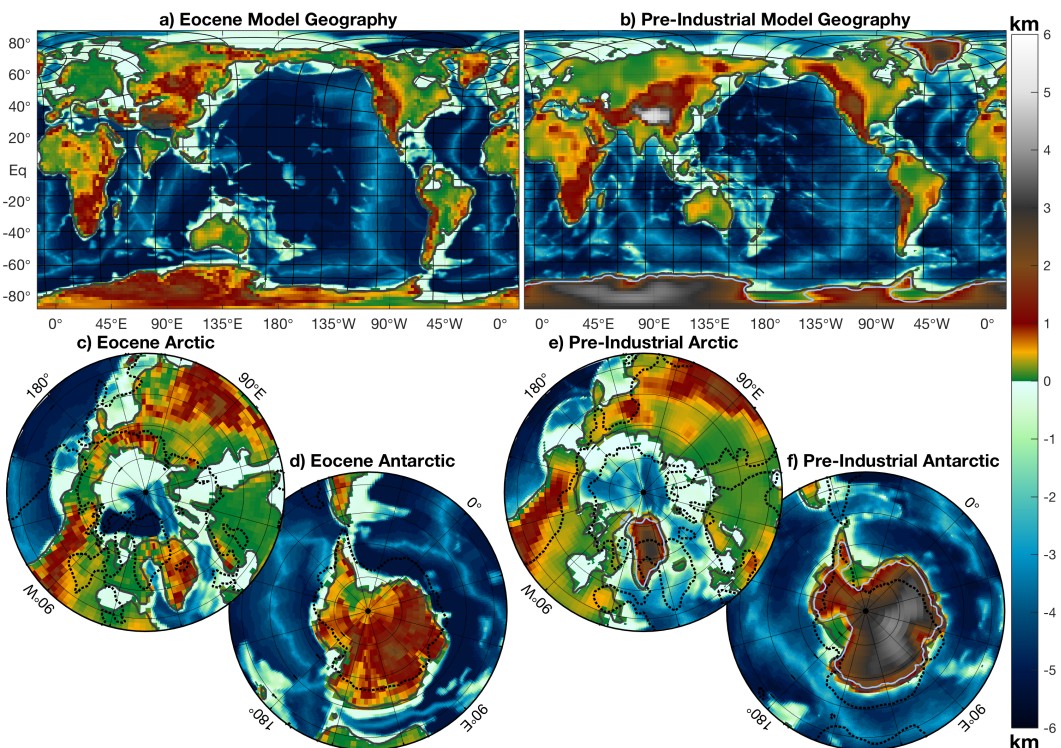

**Figure 1.** Topography (atmosphere) and bathymetry (ocean) grid used in the model simulations for the **a)** 38Ma Eocene cases and **b)** pre-industrial reference. Black lines are drawn every 20 model grid boxes for the ocean and the thick grey line shows the coastlines defined as 0.5 land fraction. Polar Stereographic projections of the Eocene model geography are shown for **c)** the Arctic and **d)** the Antarctic region, similarly for the pre-industrial one in **e)** and **f)**, respectively. Dashed black lines indicate the coastlines of the alternative geography, light blue contours show the edges of ice sheets.

To allow comparisons of various climatic features between Eocene and pre-industrial conditions within a similar framework, a pre-industrial reference run with the same version of the CESM is performed (using the geography shown in Figure 1b,e,f). Since the atmospheric component used here consists of the CAM4 at a nominal $2°$ horizontal resolution, this reference is similar to the $2°$ 1850 control of Gent et al. (2011). The solar constant in the simulation is 1361.27 W/m$^2$ and the atmospheric levels of $CO_2$ and $CH_4$ are 280ppm and 671ppb, respectively (i.e. pre-industrial carbon; PIC). Astronomical orbital parameters are set to their present-day configurations; eccentricity: 0.0167, obliquity: 23.44° and precession placing the aphelion in Northern Hemisphere summer.

Both vegetation and atmospheric aerosols are kept fixed at their respective pre-industrial distributions (Figure S1), meaning that all anthropogenic influences apart from land cover changes are disregarded. Overflow and tidal mixing parameterisations are switched off for a better comparison to the Eocene cases. Background vertical diffusivity is a constant 0.17 cm$^2$/s nearly everywhere, increasing to 0.3 cm$^2$/s around 30°N/S and 1 cm$^2$/s in the Banda Sea, following Jochum (2009).

| Case / Property | Pre-industrial | 38Ma 2×PIC | 38Ma 4×PIC |
|---|---|---|---|
| **Geography** | Present-day | 38Ma Paleomag (Baatsen et al., 2016) | |
| **Vegetation** | Pre-industrial PFTs | Eocene biomes (Sewall et al., 2000) | |
| **Aerosols** | Pre-industrial | From 50-year BAM simulation | |
| **Atmospheric CO$_2$** | 280ppm | 560ppm | 1120ppm |
| **Atmospheric CH$_4$** | 671ppb | 1342ppb | 2684ppb |
| **Spin-up length** | 3100 years | 3600 years | 4600 years |

**Table 1.** Overview of characteristics for all CESM 1.0.5 simulations that were performed (PFT: plant functional types, BAM: bulk aerosol model).

For deep-time climate simulations there are a number of model parameters and settings that need to be reconsidered, while others are left unchanged with respect to the pre-industrial reference. Apart from a newer model version and overall increases in resolution, our Eocene CESM configuration is similar to that of the CCSM3 simulations from Herold et al. (2014) and based on the NCAR suggested deep-time standards (found at: http://www.cesm.ucar.edu/models/paleo/faq). This means that overflow and tidal mixing parameterisations in the ocean are switched off and the background vertical diffusivity $\kappa_w$ is horizontally homogeneous, but dependent on model level depth $z$ as follows: $\kappa_w = vdc1 + vdc2 \tan^{-1}((|z| - dpth)linv)$, where: $vdc1 = 0.524$ cm$^2$/s, $vdc2 = 0.313$ cm$^2$/s, $dpth = 1000$m and $linv = 4.5 \cdot 10^{-3}$m$^{-1}$. The solar constant is also reduced slightly from its pre-industrial value to 1360.89 W/m$^2$ to reflect Eocene conditions. Since our Eocene model simulations are designed to reconstruct the mean climate over longer periods in time (4–8 Ma), there is not a constant set of orbital parameters that is representative for this entire time interval. A set of parameters generally conducive for Antarctic ice growth is chosen: minimum eccentricity (i.e. 0; cancelling the effect of precession) and present-day obliquity (23.44°). The low eccentricity choice is motivated by the conjunction of two such minima occurring around the EOT, as shown by Coxall et al. (2005); DeConto et al. (2008).

Our Eocene simulations use the 38Ma geography reconstruction from (Baatsen et al. 2016; Figure 1a,c,d), which in contrast to most previously used geographies is based on a palaeomagnetic ('PaleoMag') reference frame. This method prioritises on the reconstruction of exact palaeo-latitudes, crucial for palaeoclimate simulations (van Hinsbergen et al., 2015). As for any other geography reconstruction, it comes with its own limitations and uncertainties (Baatsen et al., 2016). The model

vegetation used here (Figure S1) is largely based on reconstructed biomes of Sewall et al. (2000)
and comparable to the early Eocene vegetation used by Herold et al. (2014). The mostly zonal bands
in Sewall et al. (2000) are adjusted to the new geography reconstruction and expected maritime in-
fluences, before being translated into the CLM4 plant functional types (PFTs, see also Table S1).
This translates into most of the earth being covered by various types of forests and shrubs (tropical
forest or savannah at low and mixed forests at middle-high latitudes). Neither desert regions, nor any
land ice coverage are incorporated, while elevated surfaces are covered with a separate biome. These
choices are made specifically for the middle-to-late Eocene period, for which proxy data suggests
the absence of any persistent large-scale deserts or ice sheets.

Despite being an important contribution to the radiative forcing in present and future climate simula-
tions, atmospheric aerosols are tricky to include and are therefore often either omitted or assumed to
180 be similar to pre-industrial in palaeoclimate model simulations. A considerable improvement can be
made by running a Bulk Aerosol Model (BAM, Heavens et al. 2012) version of the CAM4 to determ-
ine a more realistic distribution of aerosols. The sources of naturally formed aerosols (mainly dust,
sea salt and organic carbon, excluding volcanic emissions) are adjusted from pre-industrial levels
and redistributed based on the new land surface properties. Using these sources the (standalone)
CAM4 is run for 50 years, at the end of which a monthly climatology of aerosol distributions is
derived (see Figure S1 for the corresponding annual mean aerosol optical depth).

Two 38Ma simulations are carried out with $2 \times$PIC and $4 \times$PIC, respectively, to cover the most likely
range of atmospheric greenhouse gases during the middle-to-late Eocene ($\sim$500–1200 ppm; Beer-
ling and Royer 2011; Anagnostou et al. 2016; http://www.p-co2.org), as well as to estimate climate
sensitivity. $CO_2$ and $CH_4$ concentrations are increased simultaneously, in line with middle-to-late
Eocene estimates (Beerling et al., 2009, 2011; Goldner et al., 2014). According to Etminan et al.
(2016) the radiative forcing of $2 \times$PIC and $4 \times$PIC is equivalent to that of $2.15 \times CO_2$ and $4.69 \times CO_2$,
respectively. An overview of the different model cases analysed in this paper is given in Table 1.

### 2.3 Spin-up procedure

Our 38Ma Eocene simulations are both initialised using a stagnant ocean with a horizontally ho-
mogeneous temperature distribution, decreasing with depth: from $15^{\circ}$C at the surface to $9^{\circ}$C at the
bottom. The pre-industrial reference is initialised using present-day temperature and salinity fields
from the PHC2 dataset (Steele et al., 2001). A long spin-up (see Table 1) is performed to allow the
deep ocean to equilibrate sufficiently. An overview of absolute ($\Delta T$, $\Delta S$) and normalised ($\Delta T/T$,
$\Delta S/S$) drifts over the last 200 model years is given in Table 2 for each spin-up. Drifts are generally
$\sim 10^{-5}$–$10^{-4}$ K/year for the global average, volume weighted ocean temperature (values of $< 10^{-4}$
are often regarded as well equilibrated, see e.g. Goldner et al. 2014; Hutchinson et al. 2018). Glob-
ally averaged salinity drifts at the end of all three spin-ups are $\sim 10^{-7}$psu/year.

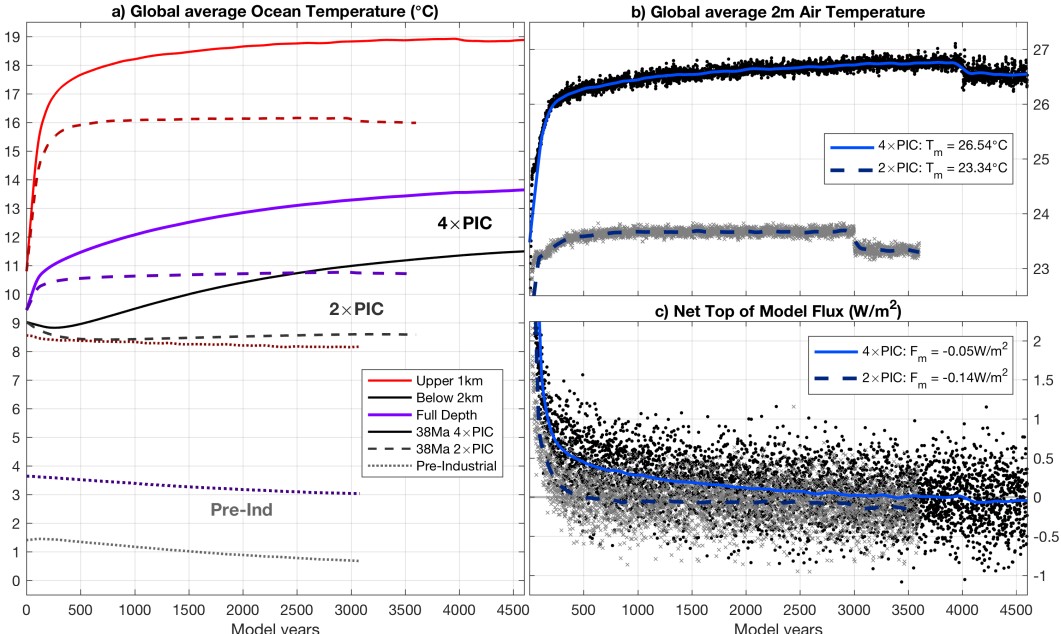

**Figure 2. a)** Time series of global upper 1km (red), below 2km (black) and full depth (purple) volume-weighted average temperature for the complete pre-industrial reference (dotted), 38Ma 2×PIC (dashed) and 4×PIC (solid) spin-up simulations. **b)** Global average near surface air temperature for the 38Ma cases (markers: annual–grey crosses: 2×PIC, black dots: 4×PIC, lines: smoothed–dashed dark blue: 2×PIC, solid light blue: 4×PIC), with mean values over the last 200 years given. **c)** Globally integrated top of model net fluxes for both 38Ma Eocene cases, using the same marker and line conventions as in b.

Time series of globally averaged upper (<1km), deep (>2km) and full depth ocean temperatures are shown for all three spin-up runs in Figure 2a. Starting from present-day initial conditions, the pre-industrial simulation cools down by about 0.5°C globally. The 38Ma 2×PIC simulation is seen to equilibrate much faster than the 4×PIC one, probably because the deep ocean equilibrium temperature is close to that of the initial state (∼9°C). With higher greenhouse gas concentrations,

the ocean experiences additional heating at the surface, causing it to become more stably stratified which consequently reduces vertical mixing into the deep ocean. This reduced mixing in the 4×PIC case causes the global average deep ocean temperature to still increase about 0.1°C over the last 500 years. As expected, changes in the upper 1km temperatures settle more quickly and are small (<0.1°C) during the last 2000 model years of each simulation.

While generally adjusting more quickly than the ocean, near surface air temperatures can also be seen to equilibrate faster in the 38Ma 2×PIC case compared to the 4×PIC one. A small but abrupt temperature drop is seen at model years 3000 and 4000 for the 2×PIC and 4×PIC case, respectively. This is caused by a regional shift in vegetation cover that was necessary to adjust regions of erroneously switched PFTs in the Eocene boundary conditions (Figure S1a shows the correct ve-

getation cover used at the end of the 38Ma simulations). Since most of the land surface is covered with different types of forest and the complexity of interactions with the modelled climate system is limited, the overall effect is minor. Both 38Ma simulations were extended by another 600 years, showing little change after a quick initial adjustment. Correcting the vegetation cover mainly affects the albedo (a $1.8 \cdot 10^{-3}$ increase globally), lowering the net fluxes at the top of the model atmosphere (with a net radiative forcing of about -0.1 W/m$^2$ and a cooling of 0.2–0.3 °C; Figure 2c). Time series of these fluxes also indicate that the atmosphere is close to radiative equilibrium towards the end of both simulations, with a $<0.1$W/m$^2$ imbalance between globally integrated shortwave and longwave fluxes. The slightly positive radiative balance agrees with the deep ocean continuing to warm up slowly while both the upper ocean and atmosphere show little change after model year $\sim$1500 for the 38Ma 4×PIC simulation.

| Simulation<br>Measure | Pre-industrial | 38Ma 2×PIC | 38Ma 4×PIC |
|---|---|---|---|
| $\Delta T$ (K/year) | $-8.7 \cdot 10^{-5}$ | $-2.6 \cdot 10^{-5}$ | $1.9 \cdot 10^{-4}$ |
| $\Delta S$ (psu/year) | $-2.2 \cdot 10^{-7}$ | $2.0 \cdot 10^{-7}$ | $-2.9 \cdot 10^{-7}$ |
| $\Delta T/T$ (/year) | $-3.2 \cdot 10^{-7}$ | $-9.1 \cdot 10^{-8}$ | $6.5 \cdot 10^{-7}$ |
| $\Delta S/S$ (/year) | $-6.2 \cdot 10^{-9}$ | $5.6 \cdot 10^{-9}$ | $-8.2 \cdot 10^{-9}$ |

**Table 2.** Overview of drifts in globally averaged (full depth, volume weighted) ocean temperature and salinity over the last 200 model years for all three CESM spin-up simulations. Normalised drifts $\Delta T/T$ and $\Delta S/S$, with $T$ temperature in Kelvin and $S$ salinity in psu, are also shown for each case using the same interval.

In addition, global patterns of ideal age tracers (a measure for oceanic ventilation timescales; Thiele and Sarmiento 1990; England 1995) are observed to equilibrate over the last 1000 years of each simulation (Supplementary Figure S2) as well as the meridional overturning strength and oceanic gateway transports (Supplementary Figure S3). Average values of ideal age still have considerable trends because of further ageing in stagnant deep ocean regions (Figure S2a) and upwelling of older water masses into the upper ocean (Figure S2b). Such changes will likely continue for many thousands of years, but the associated circulation pattern has equilibrated by the end of each simulation, which is used for further analysis. The difference in spin-up between both 38Ma cases is once again highlighted in the evolution of maximum overturning strength. Whereas the 2×PIC simulation has a stable southern overturning cell within 500 model years, its equivalent in the 4×PIC case only materialises after 2750 years (Figure S3b).

## 2.4 Proxy compilations

The 38Ma CESM simulations presented here are validated using both qualitative and quantitative measures based on proxy records of the middle-to-late Eocene. After a general assessment of the model results we look more specifically at sea surface temperatures (SSTs), where those from the 38Ma 4×PIC simulation are assumed to best represent the 42–38 Ma proxy records, and from the 2×PIC case for the 38–34 Ma interval. An overview of the considered SST proxies is given in Table S3 for 42–38 Ma and S4 for 38–34 Ma, using data from Pearson et al. (2001, 2007); Tripati et al. (2003); Kobashi et al. (2004); Bijl et al. (2009); Liu et al. (2009); Okafor et al. (2009); Douglas et al. (2014); Hines et al. (2017); Cramwinckel et al. (2018); Evans et al. (2018). A point-by-point comparison with proxy records is made, using their estimated 38Ma positions in accordance with the PaleoMag geography from Baatsen et al. (2016). Proxy-derived values for annual mean temperatures are considered with their calibration uncertainty for different methods (UK$_{37}$, TEX$_{86}^{H}$, Mg/Ca, $\Delta_{47}$ and $\delta^{18}$O). Modelled SSTs at the reconstructed proxy locations are given with error bars representing the variance in a surrounding $4° \times 4°$ box, covering the uncertainty associated with the palaeogeographic reconstruction (http://www.paleolatitude.org). Estimates using UK$_{37}$ are disregarded at low latitudes as the calibration saturates at $\sim 28°$C (Conte et al., 2006). There is an ongoing discussion on which calibration to use for TEX$_{86}$ (O'Brien et al., 2017; Cramwinckel et al., 2018; Hollis et al., 2019). In addition to SST estimates using the exponential TEX$_{86}^{H}$ calibration from Kim et al. (2010), we show those using the linear relation of Kim et al. (2008) (light blue markers in Figure 6). There is also uncertainty in the calibration of Mg/Ca, for which we use the constraints on sea water chemistry from Evans et al. (2018).

In addition to SSTs, we compare modelled near surface air temperatures to terrestrial proxies. Similarly, the distinction is made between the late middle Eocene ($\sim$42–38 Ma) and late Eocene ($\sim$38–34 Ma) as a reference for the 38Ma 4×PIC and 2×PIC simulations, respectively. The considered records are listed in Tables S5 and S6; containing data from Greenwood and Wing 1995; Gregory-Wodzicki 1997; Smith et al. 1998; Wolfe et al. 1998; Greenwood et al. 2004; Retallack et al. 2004; Hinojosa and Villagrán 2005; Uhl et al. 2007; Boyle et al. 2008; Schouten et al. 2008; Prothero 2008; Eldrett et al. 2009; Quan et al. 2012; Passchier et al. 2013). Terrestrial proxies used here consist of vegetation-based indicators using pollen, nearest living relative (NLR), leaf physiognomy (LMA, CLAMP, ELPA and LMA; Wing and Greenwood 1993; Yang et al. 2011; Traiser et al. 2005; Kowalski and Dilcher 2003) and MBT–CBT (Peterse et al., 2012).

## 2.5 Previous model results

The results of our 38Ma simulations are compared to those of Goldner et al. (2014) (hereafter: GH14) and Hutchinson et al. (2018) (H18). The latter provides an opportunity to test the robustness of the overall temperature distribution in the modelled middle-to-late Eocene cases, using very

similar boundary conditions but a different model. Here, we are mainly looking at zonally averaged sea surface and near surface air temperatures and how they correspond to the previously presented proxies. A side-by-side overview of oceanic and atmospheric fields from this study and GH14 can be found in the supplementary material.

The simulations of GH14 were carried out with the CCSM3, a predecessor of the CESM1 (the former having CAM3 rather than CAM4 with a spectral versus finite volume core, respectively). GH14 used a 45Ma Hot Spot referenced reconstruction for their geographical boundary conditions, in which Antarctica is shifted by ~6° latitude compared to the 38Ma PaleoMag reconstruction used here. They applied 4 subsequent doublings of pre-industrial $CO_2$ levels, which correspond to the EO1–EO4 cases of Cramwinckel et al. (2018). In terms of horizontal resolution GH14 implemented ~2.5° and 3.7° for the model's ocean and atmosphere grids, respectively, versus ~1° and ~2° here. A model set-up similar to ours was implemented by H18, using the 38Ma geography reconstruction from Baatsen et al. (2016), a similar oceanic resolution (~1° – atmosphere: ~3°) and a vegetation cover based on the same biome distribution used here (Figure S1a). They studied mostly the late Eocene, using the GFDL CM2.1 and subsequent doublings of present-day rather than pre-industrial $CO_2$ concentrations (i.e. ~400ppm).

A more quantitative assessment of the model's performances is made following a procedure similar to the one presented by Lunt et al. (2012), using the SST proxy compilations considered here (Tables S3 and S4). The model-proxy discrepancy $\sigma$ is defined as the average difference between model-predicted and proxy-induced SSTs at reconstructed locations (using 38Ma PaleoMag for this study and H18 versus 45Ma HotSpot for GH14). A set of different comparisons is considered, using either all proxies or a subset (i.e. global, equatorial or extra-tropical) and modelled annual mean versus summertime temperatures. An *absolute* error $|\sigma|$ is also introduced (rather than e.g. RMS) to rule out better scores by compensating errors without exaggerating the impact of a single (possibly unrealistic or not representative) proxy value.

## 2.6 Climate sensitivity

A number of additional simulations are carried out to determine the model's response to an altered radiative forcing when only the concentration of atmospheric greenhouse gases is changed. These include a 2000-year 4×$CO_2$ (Figure S16), and four 20-year perturbation experiments listed below. Using the extrapolation method introduced by Gregory et al. (2004) on the results of those shorter simulations (see Figure S17 for 4×$CO_2$), the model-derived radiative forcing from consecutive doublings of either $CO_2$ or both $CO_2$ and $CH_4$ is determined as (bracketed values: theoretical estimates from Etminan et al. 2016; E16):

- 1×$CO_2$ → 2×$CO_2$: $\Delta RF^{2 \times CO_2} = 3.49$ W/m² (E16: 3.80 W/m²).

- 1×$CO_2$ → 4×$CO_2$: $\Delta RF^{4 \times CO_2} = 7.93$ W/m² (E16: 7.96 W/m²).

- $1\times$PIC $\rightarrow$ $2\times$PIC: $\Delta RF^{2\times} = 4.18$ W/m$^2$ (E16: 4.23 W/m$^2$).

- $1\times$PIC $\rightarrow$ $4\times$PIC: $\Delta RF^{4\times} = 9.33$ W/m$^2$ (E16: 8.96 W/m$^2$).

The value of $\Delta RF^{2\times CO_2}$ determined here is slightly lower than the 3.8W/m$^2$ estimated by E16, but very close to the 3.5W/m$^2$ reported by Kay et al. (2012) using offline radiative transfer calculations for CAM4 specifically. Note that the model-derived radiative forcing is generally close to the corresponding one from E16, but featuring a stronger nonlinear behaviour towards higher perturbations especially when including the effect of CH$_4$. Using the values shown above, the radiative forcing

that results only from a second PIC doubling can thus be estimated as: $\Delta RF_{2\times}^{4\times} = 5.15$W/m$^2$ ($= \Delta RF^{4\times} - \Delta RF^{2\times}$). Through a (log scaled) quadratic fit, these estimates can also be used to show that our $2\times$PIC is equivalent to $2.25\times CO_2$ while $4\times$PIC corresponds to $4.85\times CO_2$ (rather than 2.15 and 4.69 based on E16, respectively).

Starting from the equilibrated pre-industrial reference (at model year $\sim$3100), an instant quadrupling

of atmospheric CO$_2$ was applied and continued for another 2000 model years. Using the same procedure as Gregory et al. (2004), the equilibrium climate sensitivity (ECS) of the model was assessed at 3.17°C per CO$_2$ doubling. The first 100 model years are disregarded in the extrapolation to better capture the effect of slow feedbacks (see Figure S16). Using the same atmospheric configuration and a slab ocean model, Bitz et al. (2012) reported a comparable ECS of 3.1°C in the CCSM4.

With the model-derived radiative forcing of 7.93 W/m$^2$ and an extrapolated equilibrium response of 6.34°C, the normalised equilibrium climate sensitivity of the pre-industrial reference is therefore: $S_{PI} = 0.80$°C/Wm$^{-2}$.

Since we carry out a long spin-up simulation for each of the 38Ma cases, we can determine the equi-

librium climate response as the difference between the actual equilibrium states rather than making an estimate through extrapolation. The mean over the last 200 (rather than 50 for the other results) model years of each simulation is taken to exclude as much of the internal variability as possible. When comparing global average temperature differences between the pre-industrial reference and both Eocene simulations, not all of the warming is a result of higher greenhouse gas concentrations.

To separate the *internal* effect of palaeogeography from the *externally* (i.e. greenhouse gas) driven warming, the 38Ma Eocene equilibrium climate sensitivity $S_{EO}$ can also be calculated using the combined radiative forcing due to greenhouse gas and palaeogeography changes by assuming that:

$$S_{EO} = \frac{\Delta T}{\Delta RF + G}, \tag{1}$$

where $\Delta$T is the temperature difference between two climatic states, $\Delta$RF the net radiative forcing

from greenhouse gases and $G$ the radiative forcing due to the (integral) effect of palaeogeography. Note that the latter also includes changes in the distribution of e.g. land ice and vegetation, as those are applied boundary conditions in the model set-up used here. In order to be compatible with $S_{PI}$, this formulation returns $S$ in normalised units of [°C/Wm$^{-2}$], rather than [°C per CO$_2$ doubling] as

used by Royer et al. (2012).

Since both 38Ma Eocene simulations use the same boundary conditions (except $CO_2$/$CH_4$ concentrations), it is reasonable to assume that $S$ and $G$ estimated from the comparison with the pre-industrial case should be similar (provided that their nonlinear contribution is small). By comparing the temperatures of both Eocene runs to those of the pre-industrial reference, $G$ can then be estimated from:

$$G = \frac{\Delta T^{2\times} \cdot \Delta RF^{4\times} - \Delta T^{4\times} \cdot \Delta RF^{2\times}}{\Delta T^{4\times} - \Delta T^{2\times}}, \tag{2}$$

where $\Delta \text{T}^{4\times}$ and $\Delta \text{T}^{2\times}$ denote the temperature difference with respect to the pre-industrial climate for the 38Ma 4×PIC and 2×PIC case, respectively. Within this formulation, globally averaged near surface air temperatures can then be substituted by equatorial ($<23.5°$N/S) SSTs (with a 3/2 ratio between global and equatorial change suggested by Royer et al. 2012) or deep sea temperatures, which should both be more compatible with ECS estimates from proxy data.

# 3 Results

## 3.1 The simulated middle-to-late Eocene equilibrium climate

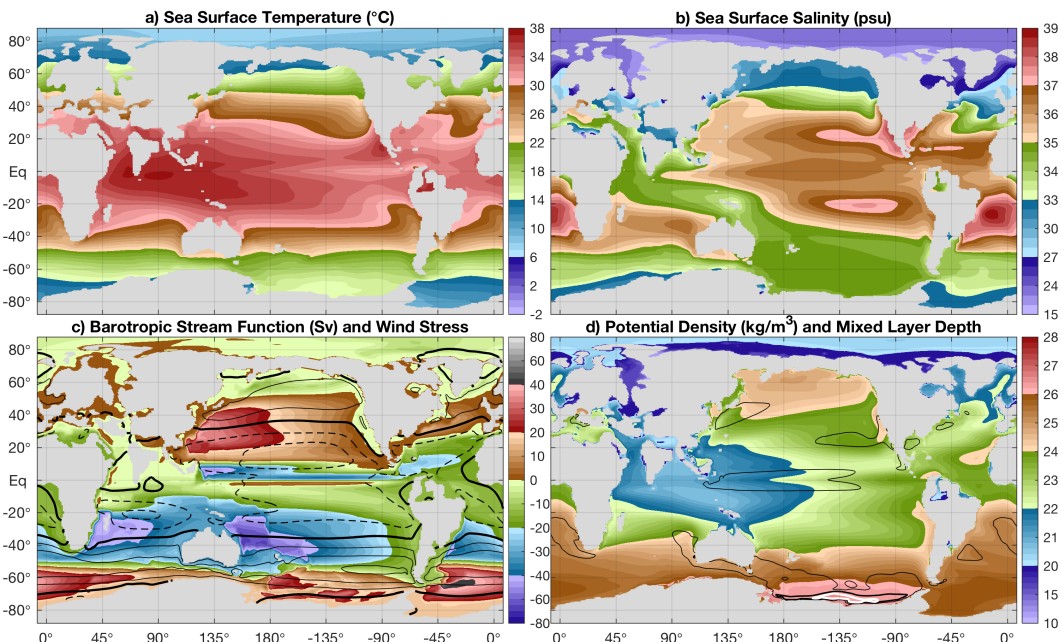

**Figure 3.** Annual mean for the 38Ma 4×PIC simulation with **a)** sea surface temperature and **b)** salinity, **c)** barotropic stream function (positive for clockwise flow) and zonal wind stress (contours every $4 \cdot 10^{-2}$Pa, solid positive and dashed negative, thick line at 0Pa), and **d)** upper 200m average potential density and mixed layer depth (contours at 100m and 250m, thick white line at 500m).

Taking the mean over the last 50 years of the 38Ma 4×PIC simulation, a number of annual mean oceanic fields are shown in Figure 3 (see also Figure S4 for the pre-industrial reference and Figure and S5 for the 38Ma 2×PIC case, seasonal fields are shown in figure S8). Sea surface temperatures are quite warm at low latitudes (34°C on average, regionally >36°C) but also mild across high latitudes (Figure 3a). Mainly the Southern Ocean and especially its Pacific part is characterised by annual mean temperatures of 10–20 °C, while those of the Arctic Ocean are mostly 6–10 °C. The former Neo-Tethys (Pacific-Indian-Mediterranean) Ocean still mostly acts as one basin in the equatorial region, with an expansive warm pool and cold tongue across its western and eastern part, respectively. Mild temperatures in the sub-polar South Pacific are accompanied by relatively high salinities of around 35psu (Figure 3b). Higher salinities are present across low latitude evaporative regions around the world, in contrast to the much fresher Arctic Ocean (∼20psu). The latter is geographically almost isolated, only connected to the shallow Para-Tethys by the Turgai Strait (<100m depth) and through the Nordic Seas (<1km depth) to the North Atlantic Ocean where consequently low surface salinities are seen as well.

The surface temperature and salinity patterns are reflected by upper 200m potential density (Figure 3d), with high values throughout the Southern Ocean and much lower densities and thus more stably stratified waters across the North Atlantic and Arctic Oceans. Deep water formation occurs only in the Pacific sector of the Southern Ocean in winter (at ~11.4C and 34.8psu), but high upper level densities suggest that this may take place virtually anywhere around the Antarctic continental slope. The only part of the Northern Hemisphere where deep water formation could occur is the high latitude Pacific Ocean, but is never seen in either of the 38Ma simulations).

The global ocean circulation consists mostly of expansive sub-tropical gyres and a geographically restricted Antarctic Circumpolar Current (ACC) and some rather weak sub-polar gyres (Figure 3c). This proto-ACC is associated with a sharp frontal zone separating warm sub-tropical from cooler sub-polar waters. Even with a shallow Drake Passage and Tasmanian Gateway (200–500 m; see Figure 1a,d), the integrated zonal flow is about 25–30 % of its pre-industrial equivalent (45–50 Sv vs 180Sv; see also Figure S3a). Strikingly, the temperature front is located at 55–60 °S, which is almost 10° poleward of where it is found in the pre-industrial reference (Figure S4). The zonal variability in the location of the front also has profound implications on regional meridional temperature contrasts. This location is strongly bound by the zonal wind stress maximum, indicating that it is fixed by the latitudes where both atmospheric and oceanic flow are the least obstructed by continents.

| Simulation<br>Measure | Pre-industrial | 38Ma 2×PIC | 38Ma 4×PIC |
|---|---|---|---|
| $\mathrm{MAT}_{glob}$ (°C) | 13.82 | 23.34 | 26.55 |
| $\mathrm{SST}_{glob}$ (°C) | 18.41 | 25.68 | 28.32 |
| $\mathrm{SST}_{eq}$ (°C) | 26.91 | 31.63 | 34.00 |
| $\mathrm{T}_{deep}$ (°C) | 0.69 | 8.59 | 11.47 |

**Table 3.** Mean equilibrium temperatures over the last 200 model years of each simulation, showing $\mathrm{MAT}_{glob}$: global average air temperature (at 2m reference height), $\mathrm{SST}_{glob}$: global average sea surface temperature, $\mathrm{SST}_{eq}$: equatorial (<23.5°N/S) average SST, and $\mathrm{T}_{deep}$: global average deep ocean ocean temperature (below 2km).

Despite differences in spin-up time, patterns of the equilibrium ocean circulation state are generally similar for our 38Ma 2×PIC and 4×PIC simulations (see also Figure S5). The annual mean, global average sea surface temperature (SST) is 28.4°C in the 4×PIC case versus 25.7°C in the 2×PIC one and 18.4°C in the pre-industrial reference (Table 3). Similar temperature differences of 2.5–3 °C are seen in both the upper and deep ocean between the 38Ma cases (Figure 2a). Globally averaged deep-sea temperatures (below 2km) reach ~11.5°C in the 38Ma 4×PIC case and ~8.6°C in the 2×PIC one, which is much warmer than for the pre-industrial ocean (~0.7°C). The difference in equatorial (<23.5°N/S) SSTs between the Eocene cases and the pre-industrial reference is smaller

(i.e. 2–3°C less than the other measures listed), but still considerable.

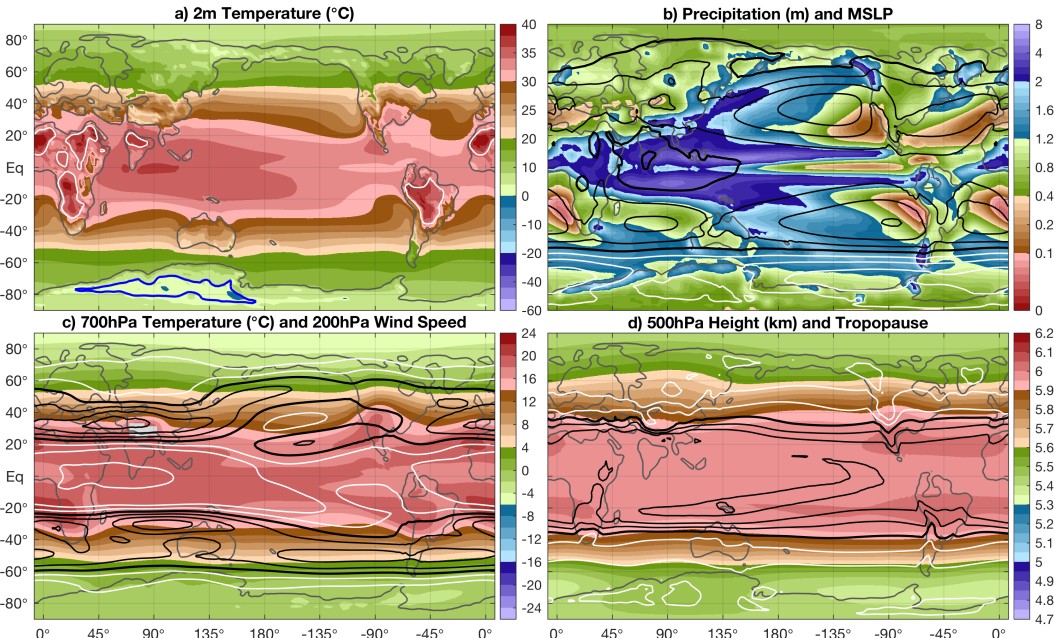

**Figure 4.** Annual mean for the 38Ma 4×PIC simulation with **a)** near surface (2m) air temperature (shading) and min/max temperature (contours; blue for $T_{min}$ <0°C and white for $T_{max}$ >40°C), **b)** precipitation (shading) and mean sea level pressure (MSLP; contours every 5hPa, thick black lines every 20hPa, ≥1000hPa in black and <1000hPa in white), **c)** 700hPa temperature (shading) and 200hPa wind speed (contours every 5m/s starting at 10m/s; white for <20m/s and black for ≥20m/s, thick black lines every 20m/s), and **d)** 500hPa geopotential height (shading) and tropopause height (contours every 1km; white for <15km and black for ≥15km, thick black line at 15km).

Also for the atmospheric fields, the mean over the last 50 years of the 38Ma 4×PIC simulation is taken and shown in Figure 4 (see also Figure S6 for the pre-industrial reference and Figure and S7 for the 38Ma 2×PIC case, seasonal fields of 4×PIC are shown in figure S9). As for the ocean, the modelled 38Ma 2×PIC atmosphere (Figure S7) is quite similar to the 4×PIC one. At 23.34°C the globally averaged, annual mean near surface air temperature at 2×PIC is ∼3.2°C cooler than at 4×PIC
(26.55°C). This is, however, still ∼9.5°C warmer than in the preindustrial reference (13.82°C). The simulated Eocene climate is overall warmer and wetter than that of the pre-industrial reference, with reduced contrasts between low and high latitudes. Near surface air temperatures (at 2m; Figure 4a) under 4×PIC are above zero and up to 40°C over land at low latitudes in the annual mean (with summer maxima regionally >50°C; see Figure S9a,b). Annual mean (daily) minimum temperatures
are only below freezing in East Antarctica and Northeast Siberia (the latter only at 2×PIC). Extreme seasonality is seen over the continental interior of Antarctica (Figure S9a,b), with 35–45 °C differ-

ences between daily mean summer and winter temperatures.

Simulated precipitation and mean sea level pressure (MSLP) patterns (Figure 4b) indicate the presence of a prominent tropical trough and expansive sub-tropical ridges in the Eocene. The inter-tropical convergence zone (ITCZ) consists of 2 precipitation maxima, extending over the Indo-Pacific basin on both sides of the equator. A pronounced double ITCZ over the Pacific Ocean is also seen in the pre-industrial reference (Figure S6b) and a known model-related issue (Song and Zhang, 2009; Bellucci et al., 2010). Effects of orographic lift are evident on westward facing coastlines and mountain ranges across middle and high latitudes (e.g. southern Andes and northern Rocky Mountains). Regardless of an overall reduced equator-to-pole temperature gradient in the 38Ma 4×PIC case, middle latitude storm tracks are prominent with increased precipitation and a poleward expansion with respect to the pre-industrial reference. Strong seasonality in the precipitation patterns also indicates the importance of monsoons in this warm Eocene climate (Figure S9c,d).

The modelled 700hPa temperature (Figure 4c) of the 38Ma 4×PIC case highlights warm mid-level air masses in persistent continental high pressure regions in the sub-tropics. The air over Antarctica is substantially warmer than over the Arctic because of its elevation and continental climate. Like the Tibetan Plateau today, the Antarctic continent therefore acts as an elevated heat island in summer (Hoskins and Karoly 1981; Ye and Wu 1998; Figure S9f). Wind speeds at 200hPa highlight the positions of both sub-tropical and polar jet streams. Several wind maxima, related to (mostly topographically induced) preferred Rossby Wave activity are evident but most pronounced at ∼90°E. A comparable jet stream pattern with regional wind maxima is seen in the pre-industrial reference, but more zonally uniform and shifted equatorward.

Reduced latitudinal as well as hemispheric differences in the modelled 38Ma 4×PIC compared to the pre-industrial reference are also seen in 500hPa geopotential and tropopause heights (Figure 4d). The latter is defined using the WMO definition of -2°C/km lapse rate. Both fields show a nearly meridionally symmetric pattern for the Eocene, with the sharpest gradient across middle-latitude regions. The overall warmer air column in the Eocene case accounts for an increase of the 500hPa surface by 100-200 m (up to 400m over Antarctica) and tropopause by 1–2 km.

We assess the global heat budget in our model results by looking at both the oceanic and atmospheric heat transports, along with their link to global circulation patterns, presented in Figure 5. The globally integrated oceanic meridional overturning stream function is similar in pattern and extent (Figure 5a) for both 38Ma Eocene simulations, but different to that of the pre-industrial reference (Figure 5b). Little change of shallow wind driven cells around the equator is seen between all three cases. A deep meridional overturning cell is present in the Southern rather than Northern Hemisphere in the 38Ma configuration, which is slightly stronger and deeper in the 2×PIC case. The effect of upwelling due to (Ekman pumping by) the zonal flow in the (proto-)ACC can be seen at around 55°S in both 38Ma cases as well as in the pre-industrial reference. The related overturning

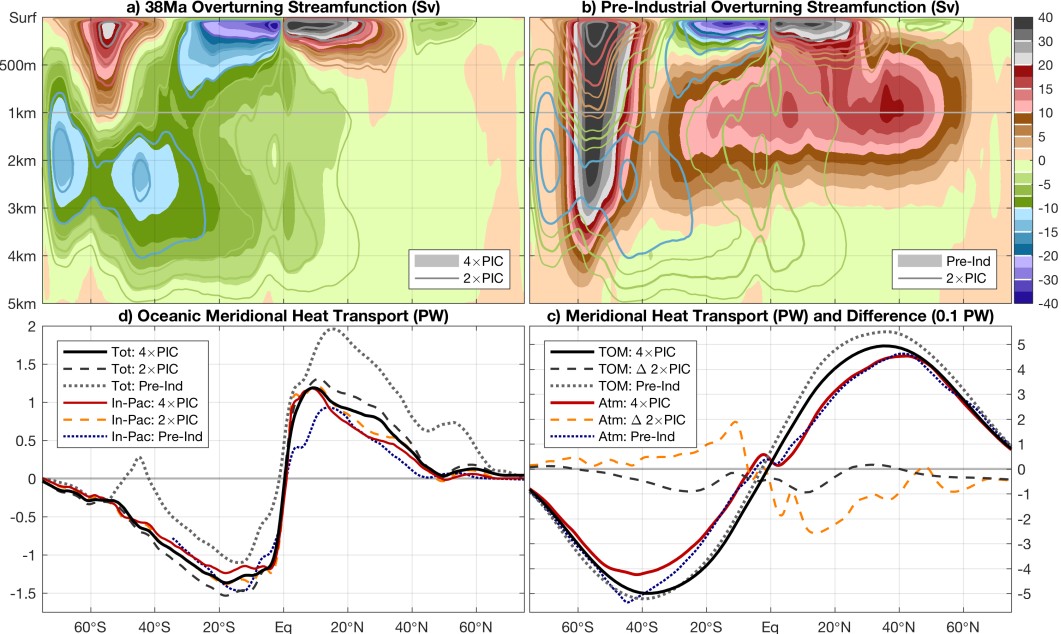

**Figure 5.** Global oceanic meridional overturning stream function, averaged over the last 50 model years of the **a)** 38Ma 4×PIC (shading) and 2×PIC (contours) simulations, and **b)** pre-industrial reference (shading; 38Ma 2×PIC again in contours). Contours are drawn using the same colour scale, at levels indicated by white lines in the colour bar to the right. **c)** Integrated meridional heat transport in the ocean, globally (greyscale) and Indo-Pacific only (coloured) for the three model cases. Note that the horizontal (latitude) scale covers the range [75°S, 75°N] with latitude increasing from left to right, for a better comparison with oceanic fields in a) and b). The top of model (TOM) required total meridional heat transport is shown in **c)** for 38Ma 4×PIC (solid black) and the pre-industrial reference (dotted grey) with the corresponding atmospheric component in red (4×PIC) and dotted blue (pre-industrial). Differences with respect to 38Ma 4×PIC of both total (dashed dark grey) and atmospheric (dashed orange) heat transport are given for the 2×PIC case, magnified tenfold.

cell in the latter is much deeper and stronger as a result of a more developed ACC but mostly the
unrestricted zonal flow in the present geographical configuration. Note that the average latitudinal
shift of these overturning cells between the modelled Eocene and pre-industrial circulation is less
pronounced than the more regional variations of the surface polar front in Figures 3 and S4.

An otherwise predominantly wind-driven gyre circulation in the 38Ma simulations is reflected by
symmetric oceanic meridional heat transport into both hemispheres (Figures 5c). This is in contrast
to the pre-industrial situation, where a difference of up to 1PW is seen between hemispheres, making
the Northern Hemisphere relatively warm (Trenberth and Caron, 2001). A major part of the 38Ma
oceanic heat transport occurs in the Neo-Tethys/Indo-Pacific Ocean, indicating the dominant role
of this basin within the Eocene circulation. The presence of a deep overturning cell in the North
Atlantic Ocean in the pre-industrial reference substantially reduces the relative Indo-Pacific contri-
bution. The largest relative differences between hemispheres are seen at high latitudes, where about

0.5PW is transported southward at 45°S while the heat transport is close to zero at 45°N in both 38Ma cases. This difference can be explained by the presence of a deep southern overturning cell, pulling warm waters into the southern high latitudes.

To determine the total required meridional heat transport, the top of model net radiative flux is integrated zonally (Figure 5d). Overall differences are small between the 38Ma cases and pre-industrial reference, with a slight shift towards the Southern Hemisphere making the Eocene pattern more symmetric. Changes seen in oceanic meridional heat transport are less pronounced in the total transport and are thus compensated by the atmospheric component, albeit only partially. This suggests that the reduced latitudinal temperature gradient seen in the modelled 38Ma cases is thus not sustained by increased meridional heat transport, but rather induces a reduced transport with respect to to the pre-industrial reference.

## 3.2 Model-proxy comparisons

In terms of the general ocean circulation, proxy-based information is limited but mostly agrees
well with what is shown by the model. Persistent deep water formation in the South Pacific Ocean
throughout most of the Eocene and the possibility of a North Pacific or South Atlantic source are
consistent with the findings of (Cramer et al., 2009; Hague et al., 2012). A strongly stratified North
Atlantic Ocean with low salinity waters at middle-high latitudes was also suggested by Coxall et al.
(2018), prior to ∼36Ma.

The all but complete absence of Arctic sea ice in the 38Ma 4×PIC model results differs from in-
dications of seasonal sea ice during the middle Eocene by Stickley et al. (2009). Although limited
in extent, the model does show sea ice during the winter months under 2×PIC (contours in Figure
S5). In addition to the sea ice indications being highly localised, age restrictions on the considered
section of the ACEX core make it difficult to rule out whether the sea ice actually occurred in the late
Eocene. Still, the model's limitations to represent sea ice need to be considered here, as the applied
scheme is quite simple. Low salinities across the Arctic Ocean would have made it easier for sea ice
to form, which is not reflected by the implemented fixed SST threshold (-1.8°C).

A southward migration of the temperature front in the Southern Ocean can help to explain some of
the high SSTs induced from proxies, especially across the Southwest Pacific region (Bijl et al., 2009;
Hines et al., 2017). Additionally, zonal variations in the front induce a thermal heterogeneity in the
Southern Ocean. The 6–8 °C (Figure 3a) difference between Tasmania and the tip of the Antarctic
Peninsula (Seymour Island) agree well with proxy indications from Douglas et al. (2014). A 10–
15° latitude difference between Tasmania and the tip of the Antarctic Peninsula in the (PaleoMag)
geography reconstruction used (Baatsen et al., 2016) here may help to explain the temperature dif-
ferences seen in proxies. On the other hand, eastward flow through the Tasmanian Gateway limits
the strength and extent of the Antarctic counter current around most of Antarctica (Bijl et al., 2013;
Houben et al., 2019). The model results disagree with indications of a strong sub-polar gyre in the
Ross Sea (limited in the model by the northern extent of East Antarctica) into the late Eocene, with
predominantly westward flow through the Tasmanian Gateway and Antarctic-derived surface wa-
ters found in the Southwest Pacific Ocean(Stickley et al., 2004; Huber et al., 2004; Bijl et al., 2013;
Cramwinckel et al., 2019). The absence of a clear temperature discrepancy between both sides of the
gateway still suggests a strong connection as seen in the model (Bijl et al., 2013), possibly explained
by atmospheric influences at shallow and near-coastal locations.

In Figure 6 a point-by-point comparison with available SST proxies (see also Table S3) is com-
plemented with zonally averaged SSTs from the model along with their longitudinal spread for both
the annual and summer mean (for 38Ma 4×PIC, see also Figure S10 for 2×PIC).

For low-latitude regions, model results and proxies are generally in good agreement (i.e. within
uncertainty). Some near-equatorial sites (e.g. Atlantic ODP Sites 925, 929 and 959) show cooler

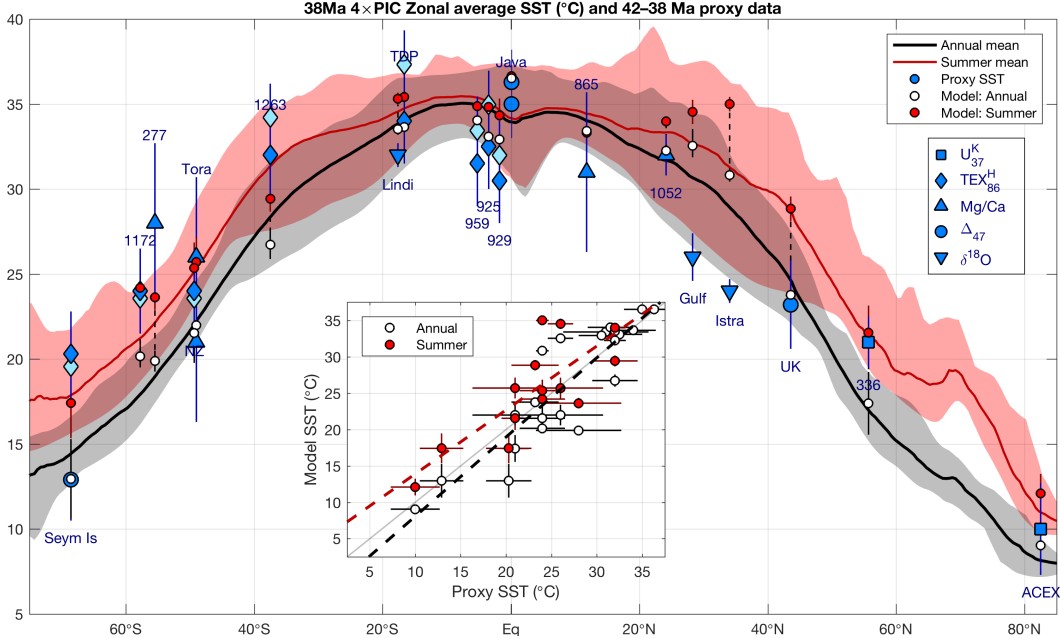

**Figure 6.** Zonal average, annual mean (black) and summer mean (red) sea surface temperature (SST) with shaded regions showing minimum and maximum values for each latitude, from the 38Ma 4×PIC simulation. Blue markers indicate estimates from 42–38 Ma SST proxies (see also Table S3; light blue: linear, dark: log calibration from Kim et al. 2008 for $\text{TEX}_{86}^H$), whereas the white (annual) and red (summer) circles depict model values at the corresponding 38Ma locations. Error bars are obtained using proxy calibration errors and the spatial variation within a $4° \times 4°$ box surrounding the corresponding location in the model. The inset shows a scatter plot comparing proxy and model SST, with dashed lines indicating a linear fit using annual mean (black) or summer mean (red; only poleward of 23°N/S) model temperatures.

temperatures than suggested by the model, at around 30°C (when considering $\text{TEX}_{86}^H$). Both the model's limitations representing the sharp upwelling region and uncertainty in the palaeolatitude can explain this offset. Cooler near-equatorial temperatures (down to <30°C) are in fact indicated by the increased zonal spread, showing that these are possible in the simulated Eocene climate. $\text{TEX}_{86}^H$ estimates from Tanzania and recent clumped isotope measurements from Java suggest that

SSTs of ~35°C or more are possible outside of upwelling regions, which is well in line with the 38Ma 4×PIC results.

A mixed agreement is seen at other latitudes, with model results seemingly too warm at northern middle latitudes, too cold at southern middle latitudes and good at high latitudes. The large discrepancy with proxy estimates from the Gulf of Mexico and Neo-Tethys (Mediterranean) stands out.

While it seems unlikely that these regions would be colder in the Eocene than they are today, the model probably underestimates strong seasonal cooling in shallow coastal waters and other localised effects near the coastline due to the complex palaeogeography at both locations. SST estimates from other locations at a similar latitude, such as ODP Site 1052 on Blake Nose (east of Florida) are much

higher bringing them back into good agreement with the model.

An improved match is generally found at most southern middle and high southern latitudes when modelled summertime temperatures are considered. Since SST proxies are based on past living organisms, they possibly have a bias towards the warm season as their activity and sedimentation are directly or indirectly affected by the available amount of sunlight (Sluijs et al., 2006, 2008; Bijl et al., 2009; Hollis et al., 2012; Schouten et al., 2013). Confidence in the modelled annual mean SSTs is

boosted by a near perfect agreement in the Arctic and clumped isotope indicators elsewhere (UK and Seymour Island).

    The modelled 38Ma 2×PIC SSTs similarly show an overall good agreement with 38–34 Ma proxy estimates, with a higher spread in the latter (Figure S10 and Table S4). The best overlap is again seen in low latitude regions, as well as a better match with modelled summer temperatures at high latit-

udes (except for clumped isotopes). Considering the model's limitations and uncertainties or possible biasses in proxy-derived SST estimates, these 38Ma simulations can reconstruct the middle-to-late Eocene (42–34 Ma) temperature distribution well.

    Despite the limited latitudinal coverage of terrestrial proxies, an overall good agreement is also

seen between the modelled temperatures on land and (Figure S11 for 38Ma 4×PIC, see also Figure S12 for 2×PIC). Proxy estimates from China, however, indicate little change in temperature while the model shows considerably warmer conditions at lower latitudes. Quan et al. (2012) discuss the limitations of capturing high summer temperatures, as it is unlikely to see similar conditions over a >20° latitude range. There are currently no proxy indications to assess the possibility of very high

model-based temperatures (exceeding 50°C, see Figure S9a,b) in low latitude continental interiors. An important limitation of the model, especially on land, is its resolution which smoothens the topography and therefore underestimates local temperature effects. A correction using the original 0.1° geography reconstruction from Baatsen et al. (2016) and a free tropospheric lapse rate of -6.5K/km significantly improves the agreement between modelled and proxy-based temperatures, especially

in North America (small squares versus filled markers in the inset of Figure S11).

### 3.3 Model-model comparisons

Zonally averaged SSTs of our 38Ma 4×PIC and 2×PIC simulations are compared to those of the 4×CO$_2$ (i.e. 1120ppm) case from GH14 and the 800ppm CO$_2$ (i.e. 2× PD) case from H18 in Figure 7, along with the pre-industrial reference and the available middle-to-late Eocene proxies. A similar overview considering near surface air temperatures and terrestrial proxies can be found in Figure 8. All of the Eocene cases feature an overall warming compared to the pre-industrial reference which is the strongest over southern high latitudes, along with a reduction of the equator-to-pole temperature gradient. Both of our 38Ma cases and the 800ppm simulation of H18 are considerably warmer than the 45Ma 4×CO$_2$ of GH14, indicating a higher sensitivity to the applied Eocene boundary conditions. It is important to mention that the simulations of H18 show deep water formation in the North Pacific Ocean, in addition to the Souther Ocean as found here and in GH14. This results in a zonal variation of northern high latitude SSTs, but little change is seen (outside of the Arctic Ocean) in zonally averaged profiles in Figure 7.

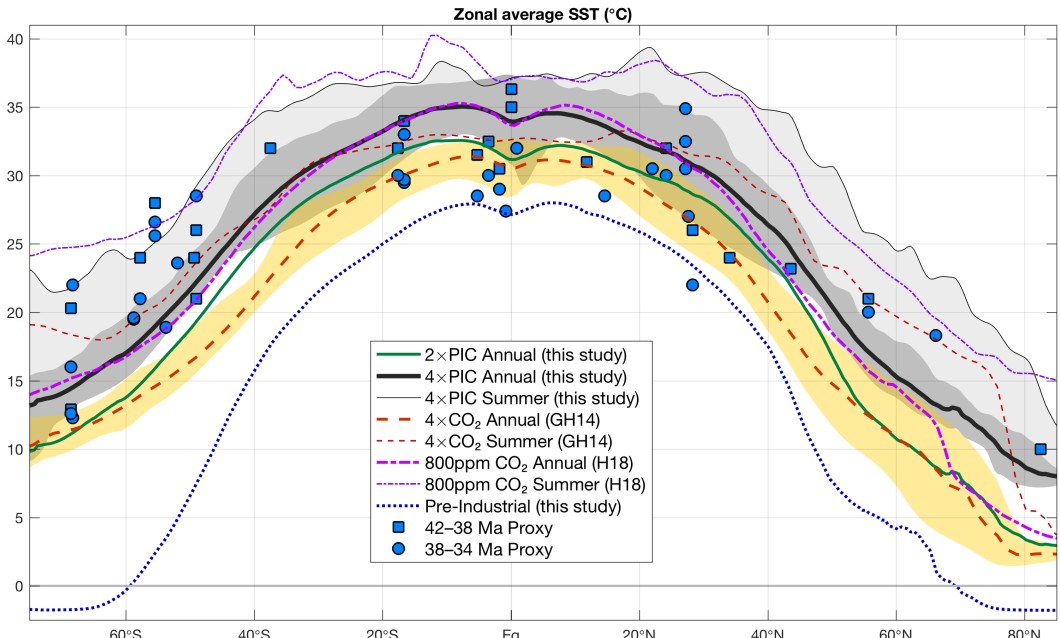

**Figure 7.** Annual mean, zonal average sea surface temperature for the 38Ma 4×PIC (black; this study) and 2×PIC (green; this study), 45Ma 4×CO$_2$ (dashed red; GH14 – Goldner et al. 2014), 38Ma 2×CO$_2$ (i.e. 800ppm, dash-dotted magenta; H18 – Hutchinson et al. 2018) and pre-industrial reference (dotted blue; this study) simulations. Estimates from proxy data are represented by blue squares for 42–38 Ma and circles for the 38–34 Ma period. Shading indicates the zonal range in temperatures for both the 38Ma 4×PIC (grey) and 45Ma 4×CO$_2$ (yellow) case, thin lines show the highest summer temperatures at each latitude (light grey shading added for 38Ma 4×PIC).

The most prominent differences in geography between the 45Ma GH14 and our 38Ma simulations are in the representation of Antarctica, the Tasmanian Gateway opening and the position of India. Especially the Southern Ocean shows large differences in circulation and the resulting temperatures, mainly related to the different model geographies. The formation of a proto-ACC and southern extent of the East Australia current act to shift the polar front in the South Pacific southward for the 38Ma cases, while the opposite happens for 45Ma (Figure S13). The southward expansion of sub-tropical gyres and migration of the associated temperature front seen here is consistent with the findings of Viebahn et al. (2016) in response to a Drake Passage closure under present-day conditions. These changes can thus be linked directly to the continental configuration and associated shifts in zonal wind stress (maximum at 55°S versus 45°S in GH14). Generally, western boundary currents (e.g. Kuroshio, Agulhas, East Australia Current) and the effects of ocean bathymetry are more pronounced in the 38Ma results. An issue in the 45Ma results with very low (negative) salinities in the Arctic Ocean, although having seemingly little impact on the general circulation, is mostly resolved (lowest salinities down to ∼10psu) in the 38Ma case by having several shallow passages.

With several indications of near-equatorial temperatures as high as 34–36 °C (Tanzania, Java and Saint Stephens Quarry), the 38Ma 4×PIC case is able to match those proxies while still allowing equatorial upwelling zones to be <30°C. The 800ppm $CO_2$ case of H18 shows similar tropical warmth, but a steeper equator-to-pole temperature gradient especially in the Northern Hemisphere. Cooler low-latitude proxies of ∼30°C are better matched by both the 45Ma 4×$CO_2$ (GH14) and 38Ma 2×PIC results. Southern Hemisphere high latitude proxies are difficult to meet by any model because of their large spread, with some of the higher estimates only matched by the warmest summertime temperatures in both the 38Ma 4×PIC and 800ppm $CO_2$ (H18) case. Still, most of the lower estimates are best reconstructed by the annual mean SSTs in the 38Ma 4×PIC case while also meeting the higher estimates when considering summer conditions.

Similar to the ocean results, all of the considered Eocene simulations show a comparable zonal average near surface air temperature distribution that differs greatly from the pre-industrial reference (Figure 8). Extreme summertime heat over low/middle latitude continental regions is seen in our 38Ma 4×PIC cases, but also present in the 38Ma 2×$CO_2$ from H18 and to a lesser extent in the 45Ma 4×$CO_2$ from GH14. Part of this warmth can be explained by the use of fixed vegetation types in all of the models, creating an efficient evaporative surface while most plants would otherwise perish. The large seasonality over Antarctica is enhanced in our 38Ma simulations compared to both GH14 and H18, with summer temperatures >30°C in the 4×PIC results (versus 20°C in GH14). This Antarctic summertime warmth, together with cloud-albedo feedbacks over the Arctic keeps annual mean, zonally averaged temperatures above 5°C in our 38Ma 4×PIC simulation across all latitudes.

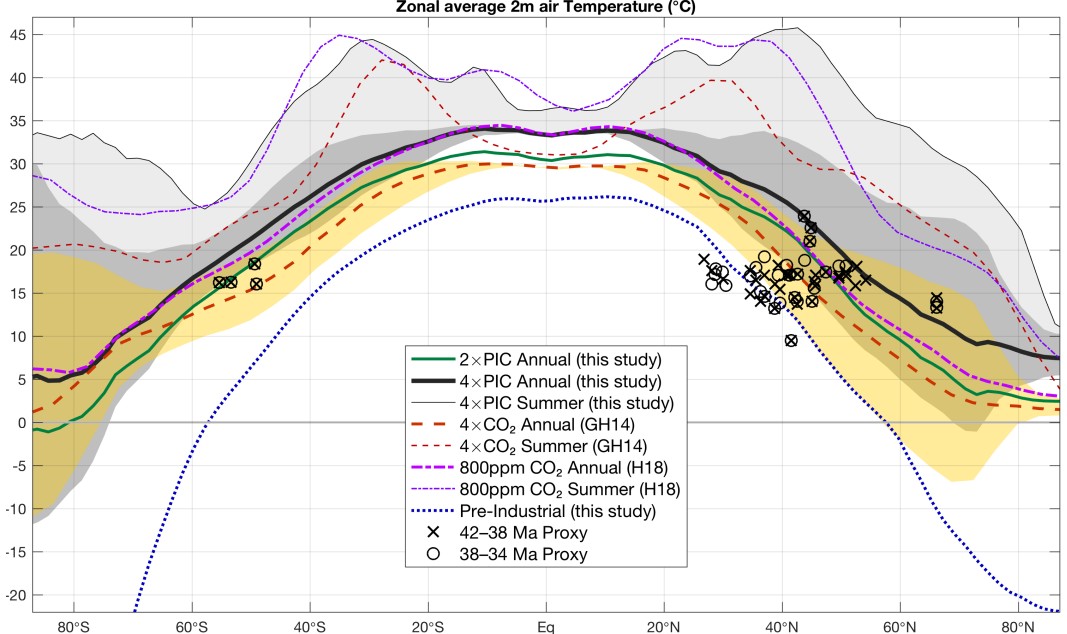

**Figure 8.** Annual mean, zonal average near-surface (2m) air temperature for the 38Ma 4×PIC (black; this study) and 2×PIC (green; this study), 45Ma 4×CO$_2$ (dashed red; GH14 – Goldner et al. 2014), 38Ma 2×CO$_2$ (i.e. 800ppm, dash-dotted magenta; H18 – Hutchinson et al. 2018) and pre-industrial reference (dotted blue; this study) simulations. Estimates from proxy data are represented by crosses for 42–38 Ma and circles for the 38–34 Ma period. Shading indicates the seasonal (DJF, JJA) range in temperatures for both the 38Ma 4×PIC (grey) and 45Ma 4×CO$_2$ (yellow) case, thin lines show the highest summer temperatures at each latitude (light grey shading added for 38Ma 4×PIC).

Global average temperatures under similar radiative forcing are strongly model-dependent for the Eocene conditions studied here. While the 45Ma 4×CO$_2$ (GH14) case is cooler than our 38Ma 2×PIC one, H18 simulate temperatures similar to those of our 4×PIC case at 800ppm CO$_2$ (albeit with a steeper equator-to-pole gradient). Although our 38Ma simulations are the only ones with elevated CH$_4$ concentrations, the difference in global temperature with GH14 is much larger than what

would be expected solely from the resulting radiative forcing. The grid and resolution used here for the atmospheric component (CAM4) are shown to both increase the sensitivity to a doubling of CO$_2$ slightly (Bitz et al., 2012). Using the CAM4 finite volume dynamical core instead of a spectral one greatly influences cloud radiative forcing and causes a warming compared to GH14 (There is a globally averaged net cloud forcing between the simulations of $13.8 - 6.4 = 7.4$W/m$^2$; see also Table

S2). Indeed, similar global temperatures are found under comparable radiative forcing by H18, who also use a finite volume dynamical core.

Part of the temperature increase between pre-industrial and Eocene conditions can be explained by the global land fraction being reduced from 29.2% to 26.2% in our CESM simulations. Lower land

fraction generally reduces the earth's albedo and therefore induces a warming response as shown by Farnsworth et al. (2019). This cannot, however, explain the lower temperatures found by GH14, who implement a further reduced land fraction of 25.1% in their model boundary conditions. A warm initialisation of the deep ocean, general circulation changes and the applied aerosol distribution add further to temperature differences between models. It is worth noting here that recent model results from Zhu et al. (2019) also found a $\sim$5°C warming in the CESM1.2 compared to their pre-industrial reference under the same external forcing, albeit for early Eocene (55Ma) conditions.

A more detailed comparison of the different components in the radiative balance between the considered simulations can be found in Table S2. The Eocene boundary conditions implemented here induce mainly a warming through the radiative forcing from greenhouse gases and a reduced albedo. The latter effect is mostly compensated by a negative cloud albedo feedback in GH14, but not in our 38Ma Eocene simulations. The warming is further enhanced by a strong positive water vapour feedback, explaining the much larger clear-sky longwave flux differences (i.e. 22.4W/m$^2$ globally) compared to what would be expected based on greenhouse gas concentrations alone.

| Simulation / Measure | 45Ma 4$\times$CO$_2$ (GH14) | 38Ma 2$\times$CO$_2$ (H18) | 38Ma 4$\times$PIC (this study) | 38Ma 2$\times$PIC (this study) |
|---|---|---|---|---|
| $\sigma_{glob}$ (°C) | $-4.81$ | $-0.91$ | $-0.43$ | $-2.49$ |
| $\sigma_{eq}$ (°C) | $-1.91$ | $+1.83$ | $+1.36$ | $+0.82$ |
| $\sigma_{ex}$ (°C) | $-6.47$ | $-2.48$ | $-1.45$ | $-4.31$ |
| $\sigma_{sum}$ (°C) | $(-2.86)\ -2.57$ | $(+0.83)\ +2.49$ | $(+0.78)\ +1.88$ | $(-0.89)\ -0.49$ |
| $\sigma_{eq} - \sigma_{ex}$ (°C) | $4.56$ | $4.31$ | $2.82$ | $5.13$ |
| $\sigma_{eq} - \sigma_{ex,sum}$ (°C) | $(+1.50)\ +1.04$ | $(+1.57)\ -1.03$ | $(+0.92)\ -0.81$ | $(+2.65)\ +2.03$ |
| $|\sigma_{glob}|$ (°C) | $5.19$ | $3.34$ | $2.84$ | $4.19$ |
| $|\sigma_{sum}|$ (°C) | $(3.35)\ 3.64$ | $(2.00)\ 3.33$ | $(1.82)\ 2.83$ | $(2.70)\ 3.06$ |
| **Average** (°C) | **4.03** | **2.75** | **2.01** | **3.31** |

**Table 4.** Model skill scores following a procedure comparable to the one presented by Lunt et al. (2012), for the model results of GH14, H18 and those presented here using the 42–38Ma SST proxy compilation from Figure 6 (see also Table S3). The last column considers the 38Ma 2$\times$PIC case and 38–34 Ma SST proxies (see Figure S10 and Table S4). Different measures are used (top to bottom); global, equatorial (<23.5° N/S), extra-tropical (>23.5° N/S), summer temperatures (only for >23.5° N/S), meridional temperature gradient (i.e. equatorial - extra-tropical; annual or summer), global absolute and summer absolute error. Bracketed values indicate model skill using modelled summer temperatures only when they improve agreement with the corresponding proxy estimate. An average skill score for each model-proxy comparison is also determined by taking the root mean square of all the above measures and highlighted in bold.

An overview of model skill scores is given in Table 4, comparing modelled SSTs from the 45Ma 4×$CO_2$ (GH14), 38Ma 800ppm (2×) $CO_2$ (H18) and our 38Ma 4×PIC cases to 42–38 Ma proxies and those of our 38Ma 2×PIC to 38–34 Ma proxies. Although all of the considered model cases simulate on average cooler SSTs than those suggested by the proxies ($\sigma_{glob} < 0$), a considerable improvement is seen for the new 38Ma cases (including H18). The equator-to-pole temperature gradient is still being overestimated by the models, but the discrepancy ($\sigma_{eq} - \sigma_{ex}$) is smallest for the 38Ma 4×PIC case presented here. As expected, better model skill is seen when considering summertime temperatures but only if those are highly underestimated by the annual mean. This is thus not the case for our 38Ma 4×PIC and H18 800ppm $CO_2$ results, as the improvement at some sites is compensated by others at which the proxies better agree with annual mean SST (see also Figures 6 and 7). Those two model cases even underestimate the meridional gradient taking summertime temperatures at extra-tropical locations (dashed red line in the inset of Figure 6). This indicates that we cannot simply assume either an overall summer-bias in those proxies or cold bias in the models at extra-tropical sites. Indeed, a further improvement of the model-proxy agreement is seen when considering modelled summer temperatures only at those sites where they provide a better match with to the corresponding proxy estimate. For nearly all of the respective skill measures the 38Ma 4×PIC results show the best agreement with the available 42–38 Ma SST proxies, followed closely by those of H18 at 800ppm $CO_2$.

### 3.4 Climate sensitivity of the middle-to-late Eocene

The difference in global average equilibrium temperature between our 38Ma 2×PIC and 4×PIC simulations (3.21°C; see Table 3) is similar to the model's response to a $CO_2$ doubling in the pre-industrial reference (3.14°C). Yet, the estimated radiative forcing from a second PIC doubling ($\Delta RF_{2\times}^{4\times} = 5.15 W/m^2$) results in a considerably lower equilibrium climate sensitivity for our Eocene simulations: $S_{EO} = 0.62°C/Wm^{-2}$.

Even in this virtually ice-free world significant polar amplification is found in the warming response to a PIC doubling, which ranges between 2°C in low latitude regions to as much as 8°C at high latitudes ($\sim$0.4–1.6 °C/Wm$^{-2}$; see Figure 9a-b). Zonal variation in the warming signal is highest at middle latitudes because of land-ocean contrasts, while differences in seasonal response are largest in polar regions. The polar amplification is reflected by differences in outgoing longwave radiation, being increasingly positive towards higher latitudes as a result of the Planck feedback (Figure 9c-d). Overall net negative longwave fluxes over equatorial regions can be explained by tropopause responses and stratospheric cooling, which reduce outgoing radiation. On a more regional scale, alternating signals of differences in shortwave and longwave radiation occur with a pronounced minimum at the equator and maxima over the ITCZ locations. This suggests the importance of cloud cover, which is seen to differ substantially (Figure 9e-f) between the 38Ma 2×PIC and 4×PIC cases. An increase in deep convective clouds over the equatorial Pacific corresponds to lower fluxes (more reflection of incoming shortwave and less outgoing longwave), while the opposite is seen off the equator. Differences in cloud cover are seen to have a warming effect over most of the high latitude regions. Increased wintertime low cloud cover over Antarctica and reduced summertime low cloud cover over the Arctic, along with increased high cloud cover all tend to warm the surface. The overall slight net heat loss (shortwave < longwave difference) at low latitudes versus a gain at middle-high latitudes agrees with a reduced meridional heat transport seen in the warmer 4×PIC climate (Figure 5c,d).

As in the present-day climate the polar amplification is due in part to albedo effects from clouds, vegetation, snow cover and some seasonal sea ice, but also the result of a strong water vapour feedback and meridionally dependent lapse rate feedback (less negative towards higher latitudes). Low latitude regions generally warm less than the global average in response to a positive radiative forcing, as they are strongly governed by moist processes (i.e. cloud cover and negative lapse rate feedback) and more tied to SSTs (low land fraction). Additionally, the total column water vapour induces a negative shortwave feedback that is strongest in the tropics and helps to explain the smaller warming between our 38Ma 2×PIC and 4×PIC than would be expected based on radiative forcing alone. An overview of zonally averaged atmospheric temperature changes between some of the different model cases considered here can be found in Figure S15.

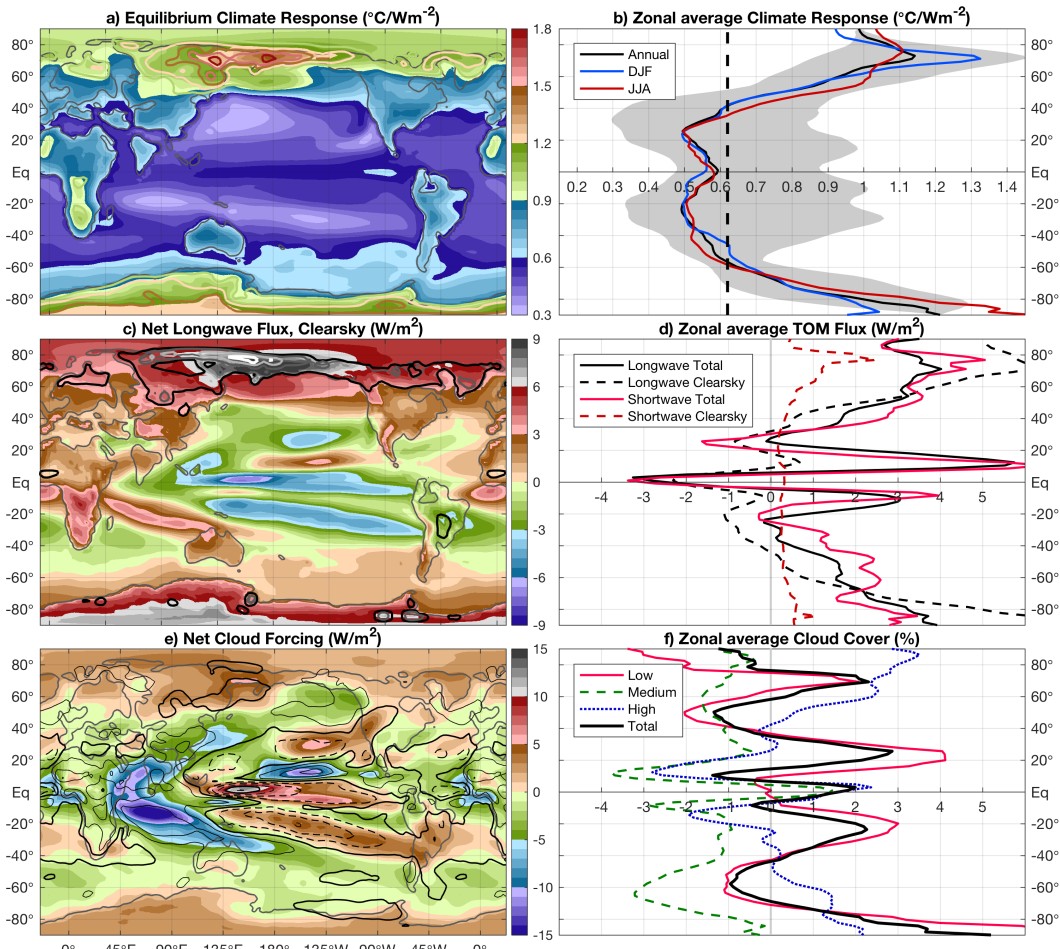

**Figure 9. a)** Annual mean temperature response, normalised per W/m$^2$ global average TOM forcing of the 38Ma 4×PIC compared to the 2×PIC equilibrium climate with contours showing the winter season only (using the same colour scale). **b)** Zonal average normalised temperature response; annual (black, shading for zonal variation), December-January-February (blue) and June-July-August (red). The (area-weighted) global average response (i.e. climate sensitivity) is $0.62°$C/Wm$^{-2}$ as indicated by the black dashed line. **c)** Clear-sky component of the net longwave flux change at the top of model (TOM) for 38Ma 4×PIC vs. 2×PIC, similar for the clear-sky shortwave flux in contours (black to white; 1, 2, 5 and 10 W/m$^2$). **d)** Zonal average TOM flux change for longwave (black) and shortwave (red) fluxes, corresponding clear-sky components are shown using dashed lines. Note that longwave fluxes are defined positive upward while shortwave fluxes are positive downward. **e)** Change in the total cloud forcing; longwave component (shading) and shortwave (contours every 5W/m$^2$; solid positive and dashed negative, thick black line at 0W/m$^2$). **f)** Zonal average changes in cloud cover (in %) at low (solid red), medium (dashed green), high (dotted blue) and all (thick solid black) atmospheric levels.

A list of averaged differences in our 38Ma 4×PIC and 2×PIC simulations with respect to the pre-industrial reference is presented in Table 5, considering global near surface air temperature ($MAT_{glob}$), equatorial SST ($SST_{eq}$) and deep ocean temperature ($T_{deep}$). Using equation 2, we can determine both the climate sensitivity $S$ and integral geography effect $G$ for the different measures. Not surprisingly, using $MAT_{glob}$ yields the same value of $S_{EO} = 0.62°C/Wm^{-2}$, with $G = 11.07 W/m^2$ (so $G \sim 6.89°C$). The estimate of $G$ decreases when using only oceanic temperatures as the direct effect of cooler temperatures over pre-industrial land ice is removed. While considering global MAT differences will likely overestimate the effect of global geography changes (mainly through lapse-rate effects on ice sheets and mountains), the opposite is true for equatorial SSTs. The according results for $S_{EO}$ show less variation between different methods as they are tied to the respective temperature changes between the 38Ma 2×PIC and 4×PIC cases. A possible range of $G = 6.08$-$11.10$ W/m$^2$ is found, equivalent to $\sim1.6$–$2.6$ $CO_2$ doublings or a 4.2–6.9°C warming globally.

| Method | $\Delta T^{4\times}$ (°C) | $\Delta T^{2\times}$ (°C) | $S_{EO}$ (°C/Wm$^{-2}$) | $G$ (W/m$^2$) |
|---|---|---|---|---|
| $MAT_{glob}$ | 12.71 | 9.514 | 0.622 | 11.10 |
| $SST_{eq}$ | 10.64 | 7.08 | 0.690 | 6.08 |
| $T_{deep}$ | 10.79 | 7.90 | 0.561 | 9.91 |

**Table 5.** Temperature differences comparing the 38Ma 4×PIC ($\Delta T^{4\times}$) and 2×PIC ($\Delta T^{2\times}$) Eocene climate to the pre-industrial reference, derived values for equilibrium climate sensitivity ($S$) and forcing from (integral) geography changes ($G$) using equation 2. Results are shown using global average 2m temperature, equatorial SST (with a 3/2 ratio, as discussed in Royer et al. 2012) and deep sea temperature (below 2km).

A more direct estimate of $G$ can be obtained by comparing a pre-industrial climate under 2×PIC and 4×PIC, with the results from the modelled 38Ma 2×PIC case. Using the model's pre-industrial ECS ($S_{PI} = 0.80°C/Wm^{-2}$) and radiative forcing of two consecutive PIC doublings ($\Delta RF^{2\times}$ and $\Delta RF^{4\times}$) yields an expected warming of 3.34°C and 7.46°C, respectively. As the modelled change in global average temperature with respect to the pre-industrial reference is either 9.50°C (2×PIC) or 12.71°C (4×PIC), an additional warming of 5.25–6.16°C takes place owing to integrated global geography changes (of which the effect is thus not entirely independent of the climatic state).

A similar exercise can be done for all of the different cases shown in Figure 10, using on one hand $S_{EO}$ and on the other hand $S_{PI}$ in combination with the previously defined values of RF and the available model results. The projected temperatures suggest a contribution of $\sim10\%$ at 2×CO$_2$/PIC and $\sim20\%$ at 4×CO$_2$/PIC from slow feedbacks (e.g. deep ocean, water vapour) to the global average warming response. The respective equilibrium temperatures for either pre-industrial or Eocene climates then yield a consistent estimate of $G = 8.6 W/m^2$, corresponding to a 5.3–6.9 °C global average temperature difference.

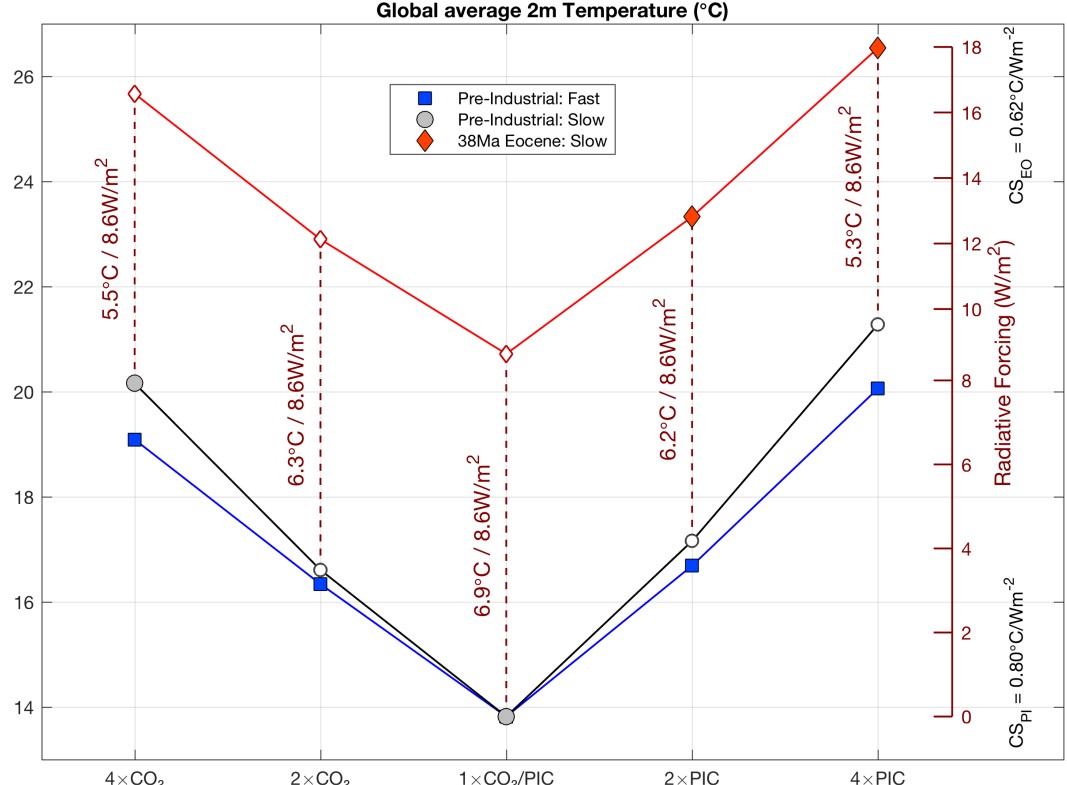

**Figure 10.** Overview of globally averaged near surface air temperatures in the different pre-industrial reference (blue) and 38Ma Eocene (red) simulations. Blue squares show the extrapolated temperature response of fast feedbacks from 20-year perturbations ($4\times CO_2$, $2\times CO_2$, $2\times PIC$ and $4\times PIC$, respectively), while grey circles also include that of slow feedbacks. Filled markers indicate results from equilibrated simulations, open markers show values that were estimated from the model-derived climate sensitivity and radiative forcing (red axis to the right; note the change in scale between pre-industrial and Eocene using $S_{PI}$ and $S_{EO}$, respectively). Offsets in both temperature and radiative forcing (i.e. $G$) between pre-industrial and 38Ma Eocene cases are specified for all of the considered atmospheric compositions.

The possible range of $G$ obtained here ($\sim$6–11W/m$^2$; 4–7°C) is considerably higher than either the estimated 1.8°C warming related to geography (using HadCM3L) found by Lunt et al. (2012) or the $\sim$2–4 W/m$^2$ (i.e. 0.6–1.1 $CO_2$ doublings) suggested by Royer et al. (2012). A comparably high warming of $\sim$5°C from integral geography effects in the Eocene was proposed by Caballero and Huber (2013) and was found more recently for the early Eocene by Zhu et al. (2019). Higher

estimates of 2–6 °C can also be deduced using the different models considered by Lunt et al. (2012).

## 4  Summary and conclusions

Using version 1.0.5 of the Community Earth System Model (CESM), we have presented the results of a simulated 38Ma Eocene climate under both high (4×PIC) and moderate (2×PIC) concentrations of $CO_2$ and $CH_4$. These are among the first simulations with a fully-coupled and detailed (CMIP5-like) climate model to study specifically the middle-to-late Eocene climate, using a recent 38Ma palaeogeography reconstruction.

The 38Ma 4×PIC case shows a warm climate with a global average near surface air temperature of 26.5°C (pre-industrial: 13.8°C) and a low equator-to-pole temperature gradient. The global heat budget is approximately meridionally symmetric, which is reflected by the zonal average temperature pattern. Deep water formation occurs in the South Pacific Ocean, while the North Atlantic is stably stratified and stagnant due to the outflow of brackish Arctic waters. A shallow and rather weak precursor of an Antarctic Circumpolar Current is found at a more southerly latitude than its present equivalent, mainly driven by the position of zonal wind stresses. Continental low/middle latitude regions are characterised by high seasonality in both hemispheres and strong summer monsoons. Middle and high latitudes mostly have mild winters, warm summers and pronounced storm tracks. The Arctic is rather cool due to its geographic isolation and the Antarctic continental interior shows strong seasonality with especially hot summers.

Comparing our 38Ma 4×PIC model results to the available 42–38 Ma sea surface and terrestrial temperature proxy records, shows an overall good match at middle-to-high latitudes without low latitudes being too hot. This indicates that the CESM is able to simulate the warm greenhouse climate of the late middle Eocene (∼Bartonian) without the need for extremely high (>1200ppm $CO_2$) greenhouse gas forcing. In the simulated 38Ma 2×PIC climate, patterns of the oceanic and atmospheric circulation are qualitatively very similar to those of the 4×PIC one. Based on a similar comparison between model results and 38–34 Ma proxy temperature estimates, our 38Ma 2×PIC case is a good analog for the late Eocene climate (∼Priabonian). While the model results are able to explain Southern Ocean heterogeneity as well as Southwest Pacific warmth suggested by proxy records, they disagree with proxy-based indications of predominant westward flow through the Tasmanian Gateway during most of the Eocene.

Previous Eocene simulations (at 4× pre-industrial $CO_2$) with a similar model but a different (45Ma) continental configuration and lower resolution resulted in overall similar sea surface temperature distributions (Goldner et al., 2014). However, our 38Ma 4×PIC case is about 4–5 °C warmer globally in both SST and land temperature. Comparable results were found by Hutchinson et al. (2018) for the late Eocene, using the same 38Ma palaeogeography reconstruction of Baatsen et al. (2016) and the GFDL CM2.1. Higher resolution and more time-specific geographic boundary conditions allow

for a better representation of regional features, including equatorial upwelling, zonal heterogeneity in the Southern Ocean and Antarctic summer warmth. Warm Eocene temperatures in our simulations compared to most previous Eocene modelling studies, are thus the combined result of a quadrupling of atmospheric methane concentrations (radiative forcing $\sim 4.85\times$ pre-industrial $CO_2$), the atmospheric dynamical core (finite volume versus spectral), different cloud parameterisations, a strong water vapour feedback and higher spatial resolution in combination with a newer palaeogeography reconstruction.

An equilibrium climate sensitivity of $S_{EO} = 0.62°C/Wm^{-2}$ is found between the 38Ma 2×PIC and 4×PIC cases, which is lower than the same model's pre-industrial value ($S_{PI} = 0.80°C/Wm^{-2}$). While some of the altered model boundary conditions for the Eocene cases add to increased temperatures with respect to the pre-industrial reference, they also act to reduce climate sensitivity (e.g. land-sea distribution and cloud cover). Water vapour and lapse rate feedbacks play crucial roles in both the reference state and sensitivity of the Eocene greenhouse climate, reducing the equator-to-pole gradient but also the overall warming response to increased atmospheric greenhouse gas concentrations. The model-derived radiative forcing from a second PIC doubling ($5.15W/m^2$) is much higher than that of a first $CO_2$ doubling ($3.49W/m^2$), starting from the pre-industrial reference. Even after 4600 model years the 38Ma 4×PIC simulation continues to warm up slightly, which suggests a possible underestimation of $S_{EO}$.

When also taking the pre-industrial reference simulation into consideration, a fixed forcing $G \approx 9W/m^2$ from (integral) geography changes in the 38Ma Eocene cases is estimated, corresponding to a $\sim 6°C$ warming globally. Previous studies have noted this effect in terms of an offset in global average temperature between pre-industrial and (pre-EOT) palaeoclimate simulations (Lunt et al., 2012; Farnsworth et al., 2019). Similar to what was found by Caballero and Huber (2013) the direct effect of ice sheet coverage is limited, leaving a considerable warming due to other geography-related changes. When using oceanic instead of atmospheric model temperatures, the influence of topography and land surface changes between the Eocene and pre-industrial cases is indeed reduced mostly by excluding the direct effect of the ice sheets and vegetation changes. Although smaller, the estimate of $G$ obtained through oceanic temperatures is still larger ($\sim 6$ W/m$^2$; 4 °C) than suggested in most previous studies. This indicates a major contribution to $G$ from changes in continental geometry, land surface properties and the related circulation patterns, which our 38Ma simulations resolve in more detail.

Several other notable (extreme) phenomena are found in our 38Ma CESM simulations. Extremely high ($\sim 50°C$) summer temperatures occur in the sub-tropics under 4×PIC and are possibly related to fixed vegetation types, but cannot be disproven by the available proxies. Strong seasonality is seen

on Antarctica, where summer temperatures reach up to 35°C in the 4×PIC case. The absence of an ice sheet, together with warm waters surrounding the continent and summertime insolation cause the Antarctic continent to become a heat island. Sea-ice coverage is nearly nonexistent and only occurs sporadically during the winter months in the 38Ma 2×PIC case.

As the simulated middle-to-late Eocene (38Ma 2×PIC and 4×PIC) climate is in good agreement with estimates from currently available proxy records, the results presented here can be used to interpret (also novel) proxy records in more detail using the modelled circulation patterns. The results mainly show that a warm and equable Eocene climate can be simulated under realistic levels of atmospheric greenhouse gases. In line with another recent study from Hutchinson et al. (2018), our 805 results can provide a basis to reconsider the conditions leading up to the EOT.

*Acknowledgements* This work was carried out under the program of the Netherlands Earth System Science Centre (NESSC), financially supported by the Dutch Ministry of Education, Culture and Science. The computations using CESM1.0.5 were done on the Cartesius at SURFsara in Amsterdam. The use of the SURFsara computing facilities was sponsored by NWO-EW (Netherlands Organisation for Scientific Research, Exact Sciences) under the project 15508. The authors would like to thank Dr. Nicholas Herold for his extensive help setting up the CESM and assisting to fix numerous bugs and difficulties emerging with deep time palaeoclimate simulations. AvdH acknowledges travel support to network partners from the EPSRC-funded Past Earth Network (Grant number EP/M008363/1) and AS thanks the European Research Council for Consolidator Grant #771497.

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
