# Peer review of "The middle-to-late Eocene greenhouse climate, modelled using the CESM 1.0.5"

_Climate of the Past, 2020_

## Referee Comment (RC1) · Anonymous Referee #1 · 10 May 2020

The middle-to-late Eocene greenhouse climate, modelled using the CESM 1.0.5 Michiel Baatsen et al.

This modelling study targets the middle to late Eocene, a time interval important to understand as it represents the close of the Cenozoic greenhouse. This is a great paper, good clear accessible explanations of model aspects, a variety of useful figures and addressing an important and outstanding issue in modelling the warmth of the early Cenozoic, i.e. how to get Antarctica as warm as proxies suggest without cranking $CO_2$ up to levels above most proxy constraints.

Modellers will likely have technical questions about the methods but my general feeling is that this paper is backed by sound theory, it uses appropriate methods and is appropriately careful in producing the data model comparison, considering different proxy

calibrations, seasonal biases etc. If anything, I think the abstract falls short of communicating some of the key findings of this paper, and its transferability to other warm climate phases, i.e. warming the poles without warming the equator too much.

My overall recommendation is publish after minor revision.

Abstract: This could better highlight some of the important tangible climate signals. It's a bit mechanical as is.

e.g.

-Highlight the gap in knowledge: i.e. What has been missing in other models, identify gap, need for different models. Remaining proxy-model mismatch at high latitudes especially. This is there in the introduction but not in the abstract.

- Emphasize that you have come some way to addressing the long-standing problem of warm poles at 2 x CO2. This is a big step forward.

-and connected . . ..emphasize that by optimizing treatment of clouds etc, and having a carefully considered and time-appropriate paleogeography you manage to warm the poles, especially Antarctica, in a way that is consistent with proxies. This has important implications for the future. . .

- Emphasize your finding of strong seasonality in the precipitation and the importance of monsoons in this warm Eocene climate

- You find variable/reduced climate sensitivity compared to today – summarize why.

Introduction and implications The paleogeography used in this model is very similar to Hutchinson et al., 2018. But you use a different model. This is thus a really good experiment opportunity to see what effects are model dependent and what are robust features. This could be emphasized better throughout. Kennedy-Asser et al., 2019 explored this idea.

Good description of the general conditions. Experiments at 4 x and 2 x modern CO2

are appropriate for the time interval.

Line 45: Add the Goldner et al., 2014 ref here.

Lines 65: add Hutchinson et al., 2018; 2019 here. Also Kennedy-Asser et al., 2019; 2020;

Line 75: The considered period is suitable to investigate both the warm greenhouse climate as the conditions leading up to the EOT. Should be and, not as?

Elsworth et al., 2017 also specifically explored the late Eocene in a model, so that's another one.

Justify why there's a need for a customized middle Eocene paleogeography between say the early Eocene (warm optimum) and late Eocene. What changes and what could make a difference?

Explain that its crucial to have different models doing the same thing to explore what features are robust between models. Kennedy-Asser et al., 2019.

Methods Model resolution; How does the model resolution compare with other models, e.g. with the Hutchinson et al., 2018 DFDS model, which professes to have a relatively high-resolution ocean? Mention this upfront. If your ocean is 1°, what kind of process should this improve upon compared to previous models?

Hutchinson et al., 2018 have proposed that the Arctic was important for some middle to EOT ocean changes. Therefore, can you add an Arctic-focused map view to fig. 1 (or SI section) to ensure its clear how this 38Ma geography treats the Arctic.

Fig. 1 caption: and corresponding text. To the caption, add where the vegetation constraints come from i.e. proxies or modelled. Worth mentioning in the caption.

"note that neither desert nor land ice are implemented"?, later on in the text the word 'incorporated' is used.

What does that mean? Do you mean that proxies and or models find no evidence for these biomes? Please clarify because this is important since any ice will have a strong albedo effect so we need be clear on this.

Some would argue that there should be small amounts of Antarctic Eocene ice. Do you think this would make a difference in your model?

Why is the BDT biome (seen in the Fig. 1c) more extensive in the northern hemisphere than on Antarctica? Is it because Antarctica is warmer than the high northern latitudes?

For the pre- industrial control – I'd like to see how the vegetation biomes are conceptualized for comparison with the 38 Ma version. Add as a supplementary figure?

More on figures:

Fig. 3 and Figs. S3 and S4, can you use the same scale increments/range and colours between the 38 Ma and PI controls -that way the differences are much clearer visually. Figure 3. Clarify in the Fig. caption that this is 38Ma. Figure 4. Explain MSLP in the caption.

LINE 380: Describes extreme seasonality on Antarctica. This is key but we are not directed to a figure/result that shows this. A reference to the supplementary figures showing seasonality at the end of this sentence would fix this.

Please keep the matching x 2 CO2 version (SI Fig S6) with the same axis temperature scale for comparison. This applies to other figure sets.

Fig. 7. This is a very useful comparison figure!

A difference between the H18 modelled 38 Ma ocean is that H18 gets Pacific overturning and you do not. This means that H18 has some northern hemisphere ocean heat transport, while you do not. Does this make a difference anywhere? Do you find compensation by the atmosphere?

Fig. 8 There is surprisingly little continental proxy data. Is it worth including data

even from a little wider time frame (early Eocene?) to get a sense of whether the temperatures on land are close to sensible for this epoch? This would be useful for the Antarctic and Arctic.

---

## Referee Comment (RC2) · Anonymous Referee #2 · 7 Jul 2020

The purpose of this work was to study the Middle-to-Late Eocene climate using a coupled model. The Middle-to-Late Eocene represents a key period of the Cenozoic characterized by the demise of the greenhouse period. The manuscript is quite long but clearly written. Its structure is logical despite some overlapping between the sections (for instance between sections 2.6 and 3.4. Moreover, these two sections have the same title). The paper relies on a large number of figures: 10 figures in the main text and 16 figures in supplementary materials. Unfortunately, the authors used a colour scale that makes the figures difficult to interpret. In addition, the superimposition of shading and contours which does not help matters. The authors do not show differences (or very occasionally) between simulations which can be very helpful (with a more classical colour scale). The authors should rewrite the abstract to better high-

light the key results of this work. Beyond this general comment, some points need to be clarified. My first comment concerns ice sheet. The authors simulate the Late Eocene climate using a pCO2 of 1120ppmv (4x) and 560ppm (2x). These values are classically used to study this period. However, the absence of ice sheet in Antarctica in the experiment at 560ppmv is more disputable. Indeed, the glaciation threshold is estimated between 560 and 920ppmv. A pCO2 as low as 560ppmv thus represents the lower limit for glaciation threshold. Moreover, it is clearly model dependent. In this work, the simulated mean annual temperature in Antarctica is below the freezing point (figure S6a), which may potentially represent required conditions for the onset of glaciation. Thus, how to be certain that an experiment without ice sheet and a pCO2 as low as 560ppmv is representative of the Priabonian period (when the CESM version 1 is used). The vegetation biomes for the Late Eocene experiments should be shown at the model resolution (Fig.1c). The cold mixed forest in the Andes seems to spread over Brazilian lowlands. The authors do not indicate how runoff was represented in the model.

L135: It can useful to better explain how the CH4 level in the Late Eocene experiments has been fixed.

L163: The distribution of aerosols is calculated using the land surface properties. Can the authors be more precise?

L190: A change in vegetation has been adjusted at the end of simulations causing a significant cooling at global scale. The explanation is not cleared. Which vegetation is shown in figure 1c?

L216: the acronym SST is used for the first time in the main text. Replace "SST" by sea surface temperature.

L245 and after (section 3.3): The authors compare their results with those of Goldner et al. (2014) and Hutchinson et al. (2018). What vegetation map were used in these two experiments? The authors argue that a lower global land fraction at Eocene induce

a lower albedo and thus a global warming. The authors should estimate the changes in earth's albedo between pre-industrial and Eocene experiments. The simulations done by Hutchinson (H18) use the same paleogeography. The only difference is the model. The authors should better explore the impact of model version.

L370: "smaller but still considerable". The authors should estimate the changes in temperature.

L380: The annually averaged (daily) minimum temperature is plotted in figure 3a. The northeastern Siberia is concerned by temperatures below the freezing point (main text) which do not appear in figure 3a.

L387: The authors should indicate where the effects of orographic lift can be observed.

L406-409: These two sentences seem to be redundant.

L465: The paleogeography of Douglas et al. (2014) is different. The difference in latitude between Tasmania and the tip of Antarctica peninsula is about 5° in Douglas' work but reaches 15° in this study. Can it explain the difference of temperature?

L467-472: How can the authors explain the absence of strong sub-polar gyre in the Ross Sea? Is it due to the paleogeography (Antarctica) or the depth of Tasmanian Gateway?

L488: The authors should indicate in table S2 and S3 where the SST proxies are located (Gulf of Mexico, Blake Nose . . .).

Minor comments: L45: reference missing : Toumoulin et al., 2020, Quantifying the effect of the Drake Passage opening on the Eocene Ocean, https://doi.org/10.1029/2020PA003889

L65: reference missing : Tardif et al., 2020, Clim. Past, https://doi.org/10.5194/cp-16-847-2020

Figure 1 caption: typo error (needleleaf)

L347: typo error (° is missing)

L432 : typo error (Indo-Pacific)

---

## Author Comment (AC1) · 4 Aug 2020

RC: This modelling study targets the middle to late Eocene, a time interval important to understand as it represents the close of the Cenozoic greenhouse. This is a great paper, good clear accessible explanations of model aspects, a variety of useful figures and addressing an important and outstanding issue in modelling the warmth of the early Cenozoic, i.e. how to get Antarctica as warm as proxies suggest without cranking $CO_2$ up to levels above most proxy constraints. Modellers will likely have technical questions about the methods but my general feeling is that this paper is backed by sound theory, it uses appropriate methods and is appro- priately careful in producing the data model comparison, considering different proxy C1 calibrations, seasonal bi- ases etc. If anything, I think the abstract falls short of com- municating some of the key

findings of this paper, and its transferability to other warm climate phases, i.e. warming the poles without warming the equator too much. My overall recommendation is publish after minor revision.

AR: The authors would like to thank the referee for the detailed review. The reviewer is overall positive and seems to agree with the main findings. Several constructive remarks are made to improve mostly on consistency and clarity, which we feel can be incorporated with mostly minor changes.

RC: Abstract: This could better highlight some of the important tangible climate signals. It's a bit mechanical as is. e.g. - Highlight the gap in knowledge: i.e. What has been missing in other models, identify gap, need for different models. Remaining proxy-model mismatch at high latitudes especially. This is there in the introduction but not in the abstract. - Emphasize that you have come some way to addressing the long-standing problem of warm poles at 2 x CO2. This is a big step forward. - and connected . . ..emphasize that by optimizing treatment of clouds etc, and having a carefully considered and time-appropriate paleogeography you manage to warm the poles, especially Antarctica, in a way that is consistent with proxies. This has important implications for the future. . . - Emphasize your finding of strong seasonality in the precipitation and the importance of monsoons in this warm Eocene climate - You find variable/reduced climate sensitivity compared to today – summarize why. Introduction and implications The paleogeography used in this model is very similar to Hutchinson et al., 2018. But you use a different model. This is thus a really good experiment opportunity to see what effects are model dependent and what are robust features. This could be emphasized better throughout. Kennedy-Asser et al., 2019 explored this idea.

AR: As the manuscript is quite long, it is challenging to keep the abstract focused while still complete. Both reviewers suggest that the abstract does not stress enough on some of the main results, so we will make the necessary changes to improve this.

RC: Introduction and implications The paleogeography used in this model is very similar to Hutchinson et al., 2018. But you use a different model. This is thus a really good experiment opportunity to see what effects are model dependent and what are robust features. This could be emphasized better throughout. Kennedy-Asser et al., 2019 explored this idea. Good description of the general conditions. Experiments at 4 x and 2 x modern CO2 are appropriate for the time interval.

AR: Considering the current length of the manuscript, it was chosen not to add a complete model inter-comparison study. Indeed, the very similar boundary conditions used by Hutchinson et al. 2018 provide a great opportunity to look at model-dependent responses. This will be stressed more when introducing the models and discussing the results, but leaving a more elaborate comparison to potential future work.

RC: Line 45: Add the Goldner et al., 2014 ref here. Lines 65: add Hutchinson et al., 2018; 2019 here. Also Kennedy-Asser et al., 2019; 2020;

AR: These references will be added.

RC: Line 75: The considered period is suitable to investigate both the warm greenhouse climate as the conditions leading up to the EOT. Should be and, not as?

AR: Indeed, this will be corrected.

RC: Elsworth et al., 2017 also specifically explored the late Eocene in a model, so that's another one.

AR: We can add this reference in the introduction as well.

RC: Justify why there's a need for a customized middle Eocene paleogeography between say the early Eocene (warm optimum) and late Eocene. What changes and what could make a difference?

AR: The palaeogeogrpahy reconstruction used here is an updated version, using more recent plate tectonic models. In addition, the reconstruction is quite detailed, which is needed to make it suitable for the model resolution used here. The timing uncertainty

in these reconstructions is considerable, so the middle Eocene time frame was chosen such that it was before most of the changes occurring around the EOT. This will be better explained and referred to in the introduction.

RC: Explain that its crucial to have different models doing the same thing to explore what features are robust between models. Kennedy-Asser et al., 2019.

AR: In line with the above comment regarding the results of Hutchinson et al. 2018, we will emphasise and consider this more throughout.

RC: Methods Model resolution; How does the model resolution compare with other models, e.g. with the Hutchinson et al., 2018 DFDS model, which professes to have a relatively high-resolution ocean? Mention this upfront. If your ocean is 1âŮ̧ę, what kind of process should this improve upon compared to previous models?

AR: The ocean resolution is similar at ∼1deg, the atmosphere has a slightly higher resolution compared to Hutchinson et al. 2018. This should lead to generally better resolved atmospheric eddies, oceanic boundary currents and gateway flows (see e.g. Bitz et al. 2012).

RC: Hutchinson et al., 2018 have proposed that the Arctic was important for some middle to EOT ocean changes. Therefore, can you add an Arctic-focused map view to fig. 1 (or SI section) to ensure its clear how this 38Ma geography treats the Arctic.

AR: To make this figure better focused on the model geography, we propose to add polar projections here and rather show the vegetation (for both Eocene and Pre-Industrial experiments) as a supplementary figure.

RC: Fig. 1 caption: and corresponding text. To the caption, add where the vegetation constraints come from i.e. proxies or modelled. Worth mentioning in the caption.

AR: This can indeed be added to the caption of the proposed supplementary figure.

RC: "note that neither desert nor land ice are implemented"?, later on in the text the

word 'incorporated' is used. What does that mean? Do you mean that proxies and or models find no evidence for these biomes? Please clarify because this is important since any ice will have a strong albedo effect so we need be clear on this.

AR: This is based on proxy reconstructions (mostly Sewall et al. 2000) and will be mentioned here.

RC: Some would argue that there should be small amounts of Antarctic Eocene ice. Do you think this would make a difference in your model?

AR: Any ice in the modelled climate would exist solely on the highest Antarctic mountains, which is unlikely to have a significant impact on the results shown here. No model simulations were carried out to test this, so we can only argue but not show this.

RC: Why is the BDT biome (seen in the Fig. 1c) more extensive in the northern hemisphere than on Antarctica? Is it because Antarctica is warmer than the high northern latitudes?

AR: The biomes are based on proxy-reconstructions and are thus not necessarily consistent with the modelled climate. This will be better pointed out in the methods section. The most likely reason for a different biome on Antarctica is the much stronger seasonality compared to the Arctic.

RC: For the pre- industrial control – I'd like to see how the vegetation biomes are conceptu- alized for comparison with the 38 Ma version. Add as a supplementary figure?

AR: This will be shown alongside the Eocene vegetation in a supplementary figure.

RC: More on figures: Fig. 3 and Figs. S3 and S4, can you use the same scale increments/range and colours between the 38 Ma and PI controls -that way the differences are much clearer visually. Figure 3. Clarify in the Fig. caption that this is 38Ma. Figure 4. Explain MSLP in the caption.

AR: The colour scales and contours are consistent between the different figures as much as possible. Yet, the differences in temperature and circulation between Eocene and pre-industrial are to such an extent that this would make a significant part of the colour scale unused. It was chosen to have the 38Ma 2x and 4x figures all using the same scales and the pre-industrial ones deviating in just a few cases. We will adjust the captions as suggested and highlight more where different conventions are used.

RC: LINE 380: Describes extreme seasonality on Antarctica. This is key but we are not directed to a figure/result that shows this. A reference to the supplementary figures showing seasonality at the end of this sentence would fix this.

AR: We should indeed refer to the figure in the supplementary material here (S8 and S10).

RC: Please keep the matching x 2 CO2 version (SI Fig S6) with the same axis temperature scale for comparison. This applies to other figure sets.

AR: Figure S6 uses the same scale as Figure 4, but deviates from Figure S5 as the latter again shows pre-industrial fields. Again, the argument here is that the latter differs so much from the Eocene cases that a consistency of scales would reduce the readability of the figures.

RC: Fig. 7. This is a very useful comparison figure! A difference between the H18 modelled 38 Ma ocean is that H18 gets Pacific overturning and you do not. This means that H18 has some northern hemisphere ocean heat transport, while you do not. Does this make a difference anywhere? Do you find compensation by the atmosphere?

AR: H18 indeed have North Pacific overturning while we do not, which is worth mentioning in the discussion. We did not make a direct comparison of oceanic heat transport, but there are no significant differences in zonally averaged temperatures. This indeed suggests that any differences in oceanic heat transport on a global change are probably compensated by the atmosphere.

RC: Fig. 8 There is surprisingly little continental proxy data. Is it worth including data even from a little wider time frame (early Eocene?) to get a sense of whether the temperatures on land are close to sensible for this epoch? This would be useful for the Antarctic and Arctic.

AR: Even stretching the considered period does not add any available terrestrial proxies to compare to either the low or high latitude regions.

---

## Author Comment (AC2) · 4 Aug 2020

RC: The purpose of this work was to study the Middle-to-Late Eocene climate using a coupled model. The Middle-to-Late Eocene represents a key period of the Cenozoic characterized by the demise of the greenhouse period. The manuscript is quite long but clearly written. Its structure is logical despite some overlapping between the sections (for instance between sections 2.6 and 3.4. Moreover, these two sections have the same title). The paper relies on a large number of figures: 10 figures in the main text and 16 figures in supplementary materials. Unfortunately, the authors used a colour scale that makes the figures difficult to interpret. In addition, the superimposition of shading and contours which does not help matters. The authors do not show differences (or very occasionally) between simulations which can be very helpful (with a

more classical colour scale). The authors should rewrite the abstract to better highlight the key results of this work.

AR: The authors want to thank the referee for this in-depth review. Both referees consistently seem to agree with the general findings and presentation of the results, pointing out that particularly the abstract needs to be re-written and improvements can be made on the conventions (regarding colours, scales and contours) used in some of the figures.

RC: Beyond this general comment, some points need to be clarified. My first comment concerns ice sheet. The authors simulate the Late Eocene climate using a pCO2 of 1120ppmv (4x) and 560ppm (2x). These values are classically used to study this period. However, the absence of ice sheet in Antarctica in the experiment at 560ppmv is more disputable. Indeed, the glaciation threshold is estimated between 560 and 920ppmv. A pCO2 as low as 560ppmv thus represents the lower limit for glaciation threshold. Moreover, it is clearly model dependent. In this work, the simulated mean annual temperature in Antarctica is below the freezing point (figure S6a), which may potentially represent required conditions for the onset of glaciation. Thus, how to be certain that an experiment without ice sheet and a pCO2 as low as 560ppmv is representative of the Priabonian period (when the CESM version 1 is used).

AR: The 560ppm experiment is indeed not a priori suitable to be carried out with a completely absent Antarctic Ice Sheet. The main reason to keep the boundary conditions consistent is to allow for a straightforward analysis of climate sensitivity under these conditions. From the results, it can actually be obtained that no ice would grow even at 560ppm but this can be pointed out beforehand to clarify the choices made (potential melt during the warm season still greatly outweighs any frozen precipitation from the cold season). Note that this does not mean that a climate with an AIS cannot exist at 560ppm, but the results shown here are still consistent with a largely ice-free Antarctica.

RC: The vegetation biomes for the Late Eocene experiments should be shown at the model resolution (Fig.1c). The cold mixed forest in the Andes seems to spread over Brazilian lowlands. The authors do not indicate how runoff was represented in the model.

AR: The vegetation biomes will be shown at model resolution along with those of the pre-industrial simulation in a supplementary figure. A brief discussion on the treatment of run-off will be added to the methods section.

RC: L135: It can useful to better explain how the CH4 level in the Late Eocene experiments has been fixed.

AR: The choice to take 2x/4x pre-industrial CH4 was based on the fact that these levels are at least as uncertain as those of CO2. The range of values taken here are in agreement with what is suggested by Beerling et al. 2009. This will be better pointed out here.

RC: L163: The distribution of aerosols is calculated using the land surface properties. Can the authors be more precise?

AR: The aerosol distributions are determined using a bulk aerosol model and is consistent with the method used in earlier studies with CCSM3/4. This will be better explained, referring to the relevant literature.

RC: L190: A change in vegetation has been adjusted at the end of simulations causing a significant cooling at global scale. The explanation is not cleared. Which vegetation is shown in figure 1c?

AR: We discovered an issue with the translation of biomes into plant functional types, causing a mix-up between several types of forest. This mainly has an albedo effect due the possibility of snow on vegetation not being implemented correctly. Adjusting the vegetation thus led to a slight cooling along with an increased surface albedo. As this effects only temperatures near the surface over land, the model response to altered

vegetation happens quickly. The biomes shown in Figure 1c are thus consistent with the corrected vegetation at the end of the simulations.

RC: L216: the acronym SST is used for the first time in the main text. Replace "SST" by sea surface temperature.

AR: This will be corrected.

RC: L245 and after (section 3.3): The authors compare their results with those of Goldner et al. (2014) and Hutchinson et al. (2018). What vegetation map were used in these two experiments? The authors argue that a lower global land fraction at Eocene induce a lower albedo and thus a global warming. The authors should estimate the changes in earth's albedo between pre-industrial and Eocene experiments. The simulations done by Hutchinson (H18) use the same paleogeography. The only difference is the model. The authors should better explore the impact of model version.

AR: The simulations of Hutchinson et al. 2018 used the same model geography and vegetation biomes, while Goldner et al. 2014 used an earlier version of the same model, a different model geography but a similar version as well. How the geography and vegetation are implemented in the models can be different, but the latter should be comparable. This will be mentioned here. An extensive overview of the radiative responses is shown in Table S1, this will be referred to more clearly. The aim here was not to provide a comprehensive comparison between all the available middle/late Eocene model studies, but rather put the results here into perspective. While we agree that it can be quite useful to do a much more in depth comparison, we would rather leave this out of the scope of this study. This motivation will be better clarified in the methods section.

RC: L370: "smaller but still considerable". The authors should estimate the changes in temperature.

AR: All of the related values are in Table 3, but this paragraph will be re-written to make

it more consistent.

RC: L380: The annually averaged (daily) minimum temperature is plotted in figure 3a. The northeastern Siberia is concerned by temperatures below the freezing point (main text) which do not appear in figure 3a.

AR: Indeed, annually averaged minimum temperature is only <0 over Siberia in the 38Ma 2xPIC case. This will be corrected here.

RC: L387: The authors should indicate where the effects of orographic lift can be observed.

AR: A few examples will be added here; e.g. North/South American middle latitudes.

RC: L406-409: These two sentences seem to be redundant.

AR: The second part of this sentence can indeed be removed.

RC: L465: The paleogeography of Douglas et al. (2014) is different. The difference in latitude between Tasmania and the tip of Antarctica peninsula is about 5degC in Douglas' work but reaches 15degC in this study. Can it explain the difference of temperature?

AR: The main difference is that the palaeogeography used here includes the effect of true polar wander, shifting some of these gateways north or south by as much as 5deg. This can indeed explain to a large extent be explained by shifts in latitude, but are also partly the result of induced circulation changes. Some discussion can be added here.

RC: L467-472: How can the authors explain the absence of strong sub-polar gyre in the Ross Sea? Is it due to the paleogeography (Antarctica) or the depth of Tasmanian Gateway? The bathymetry, overturning regime and latitude of the Ross Sea all add to the gyre strength.

AR: The possible discrepancy with proxy indications is mentioned here, but not considered further as it is yet mostly unclear on how to interpret these proxies in terms of

ocean circulation. A short discussion will be added here.

RC: L488: The authors should indicate in table S2 and S3 where the SST proxies are located (Gulf of Mexico, Blake Nose . . .).

AR: Not adding the site locations to the tables was a specific choice made to keep these reasonably compact. In addition, we will provide the original (MS Excel) files with a more complete overview of proxy site information.

RC: Minor comments: L45: reference missing : Toumoulin et al., 2020, Quantifying the effect of the Drake Passage opening on the Eocene Ocean, https://doi.org/10.1029/2020PA003889 L65: reference missing : Tardif et al., 2020, Clim. Past, https://doi.org/10.5194/cp-16- 847-2020

AR: We will add these references.

RC: Figure 1 caption: typo error (needleleaf) L347: typo error ('deg' is missing) L432 : typo error (Indo-Pacific)

AR: These will be corrected.

―――――――――――――――――――――――

---

## Author Response (AR1)

**Author's Response to: Anonymous Referee #1**

**RC**: This modelling study targets the middle to late Eocene, a time interval important to understand as it represents the close of the Cenozoic greenhouse. This is a great paper, good clear accessible explanations of model aspects, a variety of useful figures and addressing an important and outstanding issue in modelling the warmth of the early Cenozoic, i.e. how to get Antarctica as warm as proxies suggest without cranking CO2 up to levels above most proxy constraints.
Modellers will likely have technical questions about the methods but my general feeling is that this paper is backed by sound theory, it uses appropriate methods and is appropriately careful in producing the data model comparison, considering different proxy calibrations, seasonal biases etc. If anything, I think the abstract falls short of communicating some of the key findings of this paper, and its transferability to other warm climate phases, i.e. warming the poles without warming the equator too much.
My overall recommendation is publish after minor revision.

**AR**: The authors would like to thank the referee for the detailed review. The reviewer is overall positive and seems to agree with the main findings.
Several constructive remarks are made to improve mostly on consistency and clarity, which we feel are now incorporated into the new revised manuscript.

**RC**: Abstract: This could better highlight some of the important tangible climate signals. It's a bit mechanical as is.
e.g.
- Highlight the gap in knowledge: i.e. What has been missing in other models, identify gap, need for different models. Remaining proxy-model mismatch at high latitudes especially. This is there in the introduction but not in the abstract.
- Emphasize that you have come some way to addressing the long-standing problem of warm poles at 2 x CO2. This is a big step forward.
- and connected . . ..emphasize that by optimizing treatment of clouds etc, and having a carefully considered and time-appropriate paleogeography you manage to warm the poles, especially Antarctica, in a way that is consistent with proxies. This has important implications for the future. . .
- Emphasize your finding of strong seasonality in the precipitation and the importance of monsoons in this warm Eocene climate
- You find variable/reduced climate sensitivity compared to today – summarize why. Introduction and implications The paleogeography used in this model is very similar to Hutchinson et al., 2018. But you use a different model. This is thus a really good experiment opportunity to see what effects are model dependent and what are robust features. This could be emphasized better throughout. Kennedy-Asser et al., 2019 explored this idea.

**AR**: As the manuscript contains diverse material, it is challenging to keep the abstract focused while still complete.
Both reviewers suggest that the abstract does not stress enough on some of the main results, so we thoroughly re-written the abstract to improve this.

**RC**: Introduction and implications The paleogeography used in this model is very similar to Hutchinson et al., 2018. But you use a different model. This is thus a really good experiment opportunity to see what effects are model dependent and what are robust features. This could be emphasized better throughout. Kennedy-Asser et al., 2019 explored this idea. Good description of the general conditions. Experiments at 4 x and 2 x modern $CO_2$ are appropriate for the time interval.

**AR**: Indeed, the very similar boundary conditions used by Hutchinson et al. 2018 provide a great opportunity to look at model-dependent responses.
This is now better explained when introducing the models and discussing the results, but we leave a more elaborate comparison to potential future work.

**RC**: Line 45: Add the Goldner et al., 2014 ref here.
Lines 65: add Hutchinson et al., 2018; 2019 here. Also Kennedy-Asser et al., 2019; 2020;

**AR**: These references have been added.

**RC**: Line 75: The considered period is suitable to investigate both the warm greenhouse climate as the conditions leading up to the EOT. Should be and, not as?

**AR:** Indeed, this has been corrected.

**RC**: Elsworth et al., 2017 also specifically explored the late Eocene in a model, so that's another one.

**AR**: We have added this reference in the introduction as well.

**RC**: Justify why there's a need for a customized middle Eocene paleogeography between say the early Eocene (warm optimum) and late Eocene. What changes and what could make a difference?

**AR**: The palaeogeogrpahy reconstruction used here is an updated version, using more recent plate tectonic models.
In addition, the reconstruction is quite detailed, which is needed to make it suitable for the model resolution used here.
The timing uncertainty in these reconstructions is considerable, so the middle Eocene time frame was chosen such that it was before most of the tectonic changes occurring around the EOT.
This is now better explained and referred to (this motivation is also given in Baatsen et al., 2016) in the introduction.

**RC**: Explain that its crucial to have different models doing the same thing to explore what features are robust between models. Kennedy-Asser et al., 2019.

**AR**: In line with the above comment regarding the results of Hutchinson et al. 2018, is emphasised and considered more throughout.

**RC**: Methods Model resolution; How does the model resolution compare with other models, e.g. with the Hutchinson et al., 2018 DFDS model, which professes to have a relatively high-resolution ocean? Mention this upfront. If your ocean is 1°, what kind of process should this improve upon compared to previous models?

**AR**: The ocean resolution is similar at ~1deg, the atmosphere has a slightly higher resolution compared to Hutchinson et al. 2018. This should lead to generally better resolved atmospheric eddies, oceanic boundary currents and gateway flows (see e.g. Bitz et al. 2012). Some more explanation was added here.

**RC:** Hutchinson et al., 2018 have proposed that the Arctic was important for some middle to EOT ocean changes. Therefore, can you add an Arctic-focused map view to fig. 1 (or SI section) to ensure its clear how this 38Ma geography treats the Arctic.

**AR**: To make this figure better focused on the model geography, we now show only geography of both the Eocene and pre-industrial here and leave the vegetation (together with AOD) for both cases as a supplementary figure.

**RC**: Fig. 1 caption: and corresponding text. To the caption, add where the vegetation constraints come from i.e. proxies or modelled. Worth mentioning in the caption.

**AR**: This can indeed be added to the caption of the proposed supplementary figure.

RC: "note that neither desert nor land ice are implemented"?, later on in the text the word 'incorporated' is used.
What does that mean? Do you mean that proxies and or models find no evidence for these biomes? Please clarify because this is important since any ice will have a strong albedo effect so we need be clear on this.

**AR**: This is based on proxy reconstructions (mostly Sewall et al. 2000) and is now mentioned here.

**RC**: Some would argue that there should be small amounts of Antarctic Eocene ice. Do you think this would make a difference in your model?

**AR**: Any ice in the modelled climate would exist solely on the highest Antarctic mountains, which is unlikely to have a significant impact on the results shown here.
There are indeed several indications that there was at least some ice on Antarctica, mostly during the late Eocene (e.g. Scher et al. 2014, Carter et al. 2017), while the Early Eocene was most likely completely ice free.
Being in between, the middle Eocene is thus an intriguing and challenging time interval for a model study.
No experiments have been carried out in this study to specifically test the influence of ice, but leave opportunities for future work or comparison studies.

**RC**: Why is the BDT biome (seen in the Fig. 1c) more extensive in the northern hemisphere than on Antarctica? Is it because Antarctica is warmer than the high northern latitudes?

**AR**: The biomes are based on proxy-reconstructions and are thus not necessarily consistent with the modelled climate. Some more explanation on the implementation of the vegetation in the model was added, as well as a supplementary figure devoted to this. The most likely reason for a different biome on Antarctica is the much stronger seasonality compared to the Arctic.

**RC**: For the pre-industrial control – I'd like to see how the vegetation biomes are conceptualized for comparison with the 38 Ma version. Add as a supplementary figure?

**AR**: This is now shown in supplementary figure S1, together with aerosol optical depth. A supplementary table was added as well showing the relations between Eocene biomes and the plant functional types implemented in the model.

**RC**: More on figures:
Fig. 3 and Figs. S3 and S4, can you use the same scale increments/range and colours between the 38 Ma and PI controls -that way the differences are much clearer visually. Figure 3. Clarify in the Fig. caption that this is 38Ma. Figure 4. Explain MSLP in the caption.

**AR**: All of the figures in the manuscript underwent a thorough revision, adjusting the colourmaps, colour scales and contours to improve their readability as well as the consistency of similar figures. In some instances where the consistency is deliberately broken (e.g. pre-industrial barotropic stream function), it is mentioned specifically in the figure caption.

**RC**: LINE 380: Describes extreme seasonality on Antarctica. This is key but we are not directed to a figure/result that shows this. A reference to the supplementary figures showing seasonality at the end of this sentence would fix this.

**AR**: This reference was indeed missing and has been added.

**RC**: Please keep the matching x 2 $CO_2$ version (SI Fig S6) with the same axis temperature scale for comparison. This applies to other figure sets.

**AR**: These scales have been changed and matched throughout for a better comparison.

**RC**: Fig. 7. This is a very useful comparison figure!
A difference between the H18 modelled 38 Ma ocean is that H18 gets Pacific overturning and you do not. This means that H18 has some northern hemisphere ocean heat transport, while you do not. Does this make a difference anywhere? Do you find compensation by the atmosphere?

**AR**: H18 indeed have North Pacific overturning while we do not, which is now also mentioned and discussed briefly.
We did not make a direct comparison of oceanic heat transport, but there are no significant differences in zonally averaged temperatures.
This indeed suggests that any differences in oceanic heat transport on a global change are likely compensated by the atmosphere.

**RC**: Fig. 8 There is surprisingly little continental proxy data. Is it worth including data even from a little wider time frame (early Eocene?) to get a sense of whether the temperatures on land are close to sensible for this epoch? This would be useful for the Antarctic and Arctic.

**AR**: Even stretching the considered period does not add any available terrestrial proxies to compare to either the low or high latitude regions.
Regardless, taking any Early Eocene proxy records into account would not be representative for the considered period at all.
The lack of any additional terrestrial temperature proxies thus still provides great opportunities to the data community here.

**Author's Response to: Anonymous Referee #2**

**RC**: The purpose of this work was to study the Middle-to-Late Eocene climate using a coupled model.
The Middle-to-Late Eocene represents a key period of the Cenozoic characterized by the demise of the greenhouse period.
The manuscript is quite long but clearly written.
Its structure is logical despite some overlapping between the sections (for instance between sections 2.6 and 3.4. Moreover, these two sections have the same title).
The paper relies on a large number of figures: 10 figures in the main text and 16 figures in supplementary materials.
Unfortunately, the authors used a colour scale that makes the figures difficult to interpret.
In addition, the superimposition of shading and contours which does not help matters.
The authors do not show differences (or very occasionally) between simulations which can be very helpful (with a more classical colour scale).
The authors should rewrite the abstract to better highlight the key results of this work.

**AR**: The authors want to thank the referee for this in-depth review.
Both referees consistently seem to agree with the general findings and presentation of the results,
pointing out that particularly the abstract needs to be re-written and improvements can be made on the conventions (regarding colours, scales and contours) used in some of the figures. Therefore, a thorough revision of all figures was carried out, changing the colourmap, scales and contour lines to improve readability and consistency throughout. The abstract was also mostly re-written to better address the relevant findings and implications.

**RC**: Beyond this general comment, some points need to be clarified. My first comment concerns ice sheet.
The authors simulate the Late Eocene climate using a pCO2 of 1120ppmv (4x) and 560ppm (2x). These values are classically used to study this period.
However, the absence of ice sheet in Antarctica in the experiment at 560ppmv is more disputable. Indeed, the glaciation threshold is estimated between 560 and 920ppmv.
A pCO2 as low as 560ppmv thus represents the lower limit for glaciation threshold.
Moreover, it is clearly model dependent.
In this work, the simulated mean annual temperature in Antarctica is below the freezing point (figure S6a), which may potentially represent required conditions for the onset of glaciation.
Thus, how to be certain that an experiment without ice sheet and a pCO2 as low as 560ppmv is representative of the Priabonian period (when the CESM version 1 is used).

**AR**: The 560ppm experiment is indeed not a priori suitable to be carried out with a completely absent Antarctic Ice Sheet.
The main reason to keep the boundary conditions consistent is to allow for a straightforward analysis of climate sensitivity under these conditions.
From the results, it can actually be obtained that no ice would grow even at 560ppm but this can be pointed out beforehand to clarify the choices made (potential melt during the warm season still greatly outweighs any frozen precipitation from the cold season).

Note that this does not mean that a climate with an AIS cannot exist at 560ppm, but the results shown here are still consistent with a largely ice-free Antarctica.
This is now mentioned in the discussion of the results as well.

**RC:** The vegetation biomes for the Late Eocene experiments should be shown at the model resolution (Fig.1c).
The cold mixed forest in the Andes seems to spread over Brazilian lowlands.
The authors do not indicate how runoff was represented in the model.

**AR**: Some more explanation on the implementation of biomes and related plant functional types was added to the text. A new supplementary figure now also shows the model's vegetation fields and a supplementary table indicates how the Eocene biomes are converted into PFTs. A note on the treatment of runoff was added here as well.

**RC**: L135: It can useful to better explain how the CH4 level in the Late Eocene experiments has been fixed.

**AR**: The choice to take 2x/4x pre-industrial CH4 was based on the fact that these levels are at least as uncertain as those of CO2.
The range of values taken here are in agreement with what is suggested by Beerling et al. 2009.
This is now better pointed out here.

**RC**: L163: The distribution of aerosols is calculated using the land surface properties. Can the authors be more precise?

**AR**: The aerosol distributions are determined using a bulk aerosol model and is consistent with the method used in earlier studies with CCSM3/4.
This is now explained a bit more, while the resulting aerosol optical depths as they are implemented in the different simulations are shown in the new supplementary figure alongside vegetation fields.

**RC**: L190: A change in vegetation has been adjusted at the end of simulations causing a significant cooling at global scale. The explanation is not cleared. Which vegetation is shown in figure 1c?

**AR**: We discovered an issue with the translation of biomes into plant functional types, causing a mix-up between several types of forest.
This mainly has an albedo effect due the possibility of snow on vegetation not being implemented correctly.
Adjusting the vegetation thus led to a slight cooling along with an increased surface albedo.
As this effects only temperatures near the surface over land, the model response to altered vegetation happens quickly.
The biomes shown in Figure 1c are thus consistent with the corrected vegetation at the end of the simulations.
This is now mentioned here and additional explanation on the effect of changing the vegetation was added.

**RC**: L216: the acronym SST is used for the first time in the main text. Replace "SST" by sea surface temperature.

**AR**: This has been corrected.

**RC**: L245 and after (section 3.3): The authors compare their results with those of Goldner et al. (2014) and Hutchinson et al. (2018). What vegetation map were used in these two experiments?
The authors argue that a lower global land fraction at Eocene induce a lower albedo and thus a global warming.
The authors should estimate the changes in earth's albedo between pre-industrial and Eocene experiments.
The simulations done by Hutchinson (H18) use the same paleogeography. The only difference is the model.
The authors should better explore the impact of model version.

**AR**: The simulations of Hutchinson et al. 2018 used the same model geography and vegetation biomes, while Goldner et al. 2014 used an earlier version of the same model, a different model geography but a similar version as well.
How the geography and vegetation are implemented in the models can be different, but the latter should be comparable. This is now mentioned and explained better.
An extensive overview of the radiative responses is shown in Table S1, this is referred to more clearly.
The aim here was not to provide a comprehensive comparison between all the available middle/late Eocene model studies, but rather put the results here into perspective.
While we agree that it can be quite useful to do a much more in depth comparison, we would rather leave this out of the scope of this study.
This motivation is now better clarified in the methods section.

**RC**: L370: "smaller but still considerable". The authors should estimate the changes in temperature.

**AR**: All of the related values are in Table 3 but this was unclear, so the paragraph is re-written to make it more consistent.

**RC**: L380: The annually averaged (daily) minimum temperature is plotted in figure 3a. The northeastern Siberia is concerned by temperatures below the freezing point (main text) which do not appear in figure 3a.

**AR**: Indeed, annually averaged minimum temperature is only <0 over Siberia in the 38Ma 2xPIC (and not 4xPIC) case. This has been corrected.

**RC**: L387: The authors should indicate where the effects of orographic lift can be observed.

**AR**: A few examples have been added here; e.g. North/South American middle latitudes.

**RC**: L406-409: These two sentences seem to be redundant.

**AR**: A part was left twice here, the second sentence has been removed.

**RC**: L465: The paleogeography of Douglas et al. (2014) is different.
The difference in latitude between Tasmania and the tip of Antarctica peninsula is about 5◦ in Douglas' work but reaches 15◦ in this study.
Can it explain the difference of temperature?

**AR**: The main difference is that the palaeogeography used here includes the effect of true polar wander, shifting some of these gateways north or south by as much as 5deg.
This can indeed explain to a large extent be explained by shifts in latitude, but are also partly the result of induced circulation changes.
Some more discussion has been added here.

**RC**: L467-472: How can the authors explain the absence of strong sub-polar gyre in the Ross Sea?
Is it due to the paleogeography (Antarctica) or the depth of Tasmanian Gateway?
The bathymetry, overturning regime and latitude of the Ross Sea all add to the gyre strength.

**AR**: The possible discrepancy with proxy indications is mentioned here, but not considered further as it is yet mostly unclear on how to interpret these proxies in terms of ocean circulation.
A short discussion has been added here.

**RC**: L488: The authors should indicate in table S2 and S3 where the SST proxies are located (Gulf of Mexico, Blake Nose . . .).

**AR**: Not adding the site locations to the tables was a specific choice made to keep these reasonably compact.
In addition, we provide the original (MS Excel) files with a more complete overview of proxy site information.

**RC**: Minor comments: L45: reference missing : Toumoulin et al., 2020, Quantifying the effect of the Drake Passage opening on the Eocene Ocean,
https://doi.org/10.1029/2020PA003889
        L65: reference missing : Tardif et al., 2020, Clim. Past, https://doi.org/10.5194/cp-16- 847-2020

**AR**: We have added these references.

**RC**: Figure 1 caption: typo error (needleleaf)
        L347: typo error (◦ is missing)
        L432 : typo error (Indo-Pacific)

**AR**: These have been be corrected.